# UNBIASED OBJECT DETECTION BEYOND FREQUENCY WITH VISUALLY PROMPTED IMAGE SYNTHESIS

**Xinhao Cai**[1,4]\*, **Liulei Li**[2]\*, **Gensheng Pei**[3], **Tao Chen**[1,4]
**Jinshan Pan**[1], **Yazhou Yao**[1,4]†, **Wenguan Wang**[2]
[1] Nanjing University of Science and Technology    [2] Zhejiang University
[3] Department of Electrical and Computer Engineering, Sungkyunkwan University
[4] State Key Laboratory of Intelligent Manufacturing of Advanced Construction Machinery
`https://github.com/NUST-Machine-Intelligence-Laboratory/Beyond_Freq`

## ABSTRACT

This paper presents a generation-based debiasing framework for object detection. Prior debiasing methods are often limited by the representation diversity of samples, while naive generative augmentation often preserves the biases it aims to solve. Moreover, our analysis reveals that simply generating more data for rare classes is suboptimal due to two core issues: i) instance frequency is an incomplete proxy for the true data needs of a model, and ii) current layout-to-image synthesis lacks the fidelity and control to generate high-quality, complex scenes. To overcome this, we introduce the representation score (RS) to diagnose representational gaps beyond mere frequency, guiding the creation of new, unbiased layouts. To ensure high-quality synthesis, we replace ambiguous text prompts with a precise visual blueprint and employ a generative alignment strategy, which fosters communication between the detector and generator. Our method significantly narrows the performance gap for underrepresented object groups, *e.g.*, improving large/rare instances by 4.4/3.6 mAP over the baseline, and surpassing prior L2I synthesis models by 15.9 mAP for layout accuracy in generated images.

## 1 INTRODUCTION

The reliability of object detection models is fundamentally limited by biases in their training data, manifesting as skewed distributions across object categories (Ouyang et al., 2016), sizes (Herranz et al., 2016), and spatial locations (Zheng et al., 2024). Conventional debiasing strategies, such as resampling (Cui et al., 2019) or re-weighting (Tan et al., 2020), attempt to mitigate this by adjusting the influence of training instances based on frequency. While effective to a degree, these methods are **constrained by the visual vocabulary** of the original dataset. They can re-balance the influence of rare samples but cannot generate novel appearances or contexts to fill representational gaps.

Generation-based data augmentation (Wu et al., 2023; Trabucco et al., 2024) has emerged as a promising alternative to overcome this limitation. By synthesizing entirely new training samples, these methods hold the potential to create a more balanced dataset. However, current solutions for object detection typically follow a layout-to-image (L2I) synthesis pipeline (Chen et al., 2024a; Wang et al., 2024), where the layouts used as conditions for data generation are directly sampled from the original training set. Thus the generation process inevitably **preserves the very biased distributions** they aim to solve, leaving a clear need for a truly bias-aware generation strategy.

But what would an effective generation-based debiasing framework entail? Our investigation in §2 reveals that: **i)** simply combining the frequency-centric debiasing view with generative approaches, *i.e.*, generating more images for rare data groups, is not the final answer. It can outperform both traditional augmentation techniques (*e.g.*, copy-paste, random flip, crop) and bias-agnostic L2I synthesis, yet still falls short of the gains achieved by enriching the training set with more real samples; **ii)** the quality of samples generated by current L2I synthesis methods remains below that of real data, as models trained on synthetic samples consistently underperform those trained on real ones.

---

\*Equal contribution.
†Corresponding author.

The problems can be two sets: ❶ *Instance frequency is an incomplete proxy to determine the most needed data of a model* (Chawla et al., 2002; He & Garcia, 2009). According to the controlled experiments in §2, we find that certain high-performing and data-rich groups (*e.g.*, large objects) can be more 'data hungry' and gain greater benefit from additional data compared to low-performing groups with limited samples (*e.g.*, small objects). Relying solely on frequency can result in suboptimal interventions. ❷ *Even with a perfect, bias-targeted layout and a powerful generation model, current L2I approaches struggle to render new samples faithfully.* Prior L2I methods primarily focus on fusing layout conditions into the generation process, with limited attention given to enhancing the fidelity of generated images to real-world data. Moreover, these methods directly translate 2D spatial arrangements into 1D text sequences. This introduces ambiguity and lacks the fine-grained control for complex scenes with specific object relationships and occlusion (Johnson et al., 2018).

In this work, we propose a targeted debiasing framework that automatically diagnoses the underrepresented data groups and executes precise generation to diversify training data. To tackle ❶, we introduce a ***representation score*** (RS) that moves beyond simple frequency counts to quantify how well a concept is represented across both sample density and representation diversity. The RS then guides a bias-aware recalibration module which constructs new, unbiased layouts to fill the identified representational gaps. Furthermore, the entire diagnosis-then-create pipeline is embedded within a ***dynamic debiasing engine*** that leverages detector errors to continuously refine the RS, ensuring the system remains adaptive and focused on the challenging biases throughout training. To tackle ❷, we replace ambiguous text prompts with a ***visual blueprint***, *i.e.*, canvases composed of colored rectangles that specify the class, size, and position of each object. This provides the generative model with direct and unambiguous instructions on object relationships, occlusion, and instance identity, ensuring the precise synthesis of debiased samples. Next, we exploit the duality between L2I synthesis and object detection, where the output of one task naturally serves as the input to the other. On this basis, we form a ***generative alignment*** mechanism that enforces consistency within an "Image-Layout-Image" loop. This facilitates communication between the generator and detector by penalizing the detector when it produces layouts that are insufficient for faithful image synthesis.

Unlike frequency-based methods, our RS-driven debiasing strategy tackles limited sample diversity by completing the truly underrepresented data groups with samples featuring novel appearances; moving beyond conventional generative augmentation, visual-blueprint and generative alignment facilitates precise synthesis of high-quality data targeting specific representation gaps. Consequently, our method demonstrates strong debiasing effectiveness. It establishes a new SOTA and greatly narrows the performance gap for underrepresented groups, *e.g.*, **+3.6** mAP for rare classes, **+3.2** mAP for instances at image borders, **+4.4** and **1.9** mAP for large and small objects on MS COCO. Our approach also demonstrates high generation fidelity, with the accuracy of layouts in synthesized images surpassing prior SOTA by **15.9** mAP, when compared against existing L2I synthesis models.

## 2    THE FREQUENCY TRAP AND FIDELITY GAP: A MOTIVATING STUDY

In this section, we conduct controlled experiments across three dimensions: spatial location, category frequency, and object size, to assess the influence of different data augmentation and debiasing strategies. All studies utilize Faster R-CNN (Ren et al., 2015) with a ResNet-50 (He et al., 2016) backbone. Hyperparameters are kept identical across models. We first train the detector on a random 1/4 subset of the MS COCO training set. We then measure the mAP by enriching the 1/4 subset by factors of 4/3, 2, and 4 with: **i)** resampling (Gupta et al., 2019) rare data groups via standard data augmentation techniques like copy-paste, random flip, and crop (termed *Data Aug*); **ii)** bias-agnostic L2I synthesis to generate new samples using layouts from training sets (termed *Bias-Agnostic Gen*); **iii)** resampling rare data groups via L2I synthesis (termed *Freq-Aware Gen*); and **iv)** real samples from the remaining 3/4 training set (termed *Real Data*). Results are reported by $mAP_{center, middle, outer}$ for spatial location; $mAP_{frequent, common, rare}$ for object category; and $mAP_{large, normal, small}$ for object size. Detailed definitions for metrics are provided in ***Appendix***. Results are summarized in Fig. 1

● **Observation 1: Generative Debiasing Outperforms Traditional Augmentation, Yet Falls Short of a Complete Remedy.** The *Freq-Aware Gen* strategy, which uses L2I synthesis to create new instances for rare data groups, consistently outperforms the *Data Aug* baseline across all dimensions. But when compared to models trained by *Real Data*, its performance still lags behind.

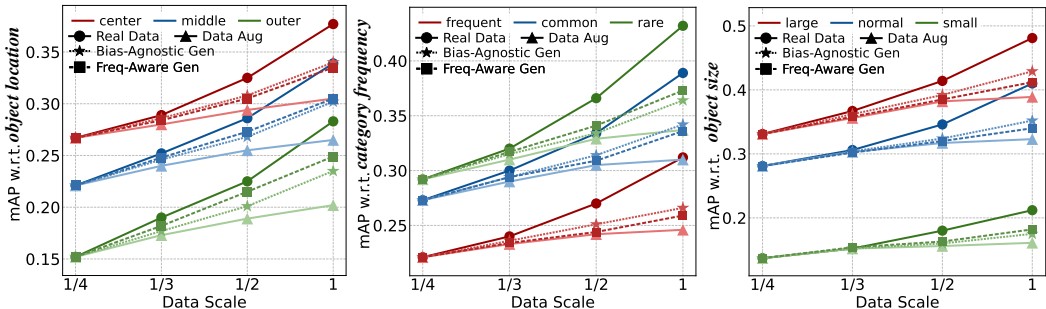

Figure 1: Comparison of four data enrichment strategies with respect to object location, category frequency, and object size as the dataset scale increases from 1/4 by factors of 4/3, 2, and 4.

ANALYSIS: These results support our claim that *Data Aug* is "constrained by the visual vocabulary of the original dataset", leading to limited diversity and improvement. The superiority of *Freq-Aware Gen* confirms that generation-based augmentation is a promising alternative. At the same time, its failure to match *Real Data* proves that current solutions are not the final answer.

• **Observation 2: Frequency is an Incomplete and Potentially Misleading Proxy for Data Need.** We observed that certain data-rich groups, such as large objects, benefited disproportionately more from additional samples in *Bias-Agnostic Gen* (+9.8 mAP) than *Freq-Aware Gen* (+8.1 mAP). This indicates that relying merely on frequency can lead to a suboptimal intervention.

ANALYSIS: This provides direct evidence for our claim that "Instance frequency is an incomplete proxy to determine the most needed data of a model", and "Relying solely on frequency can result in suboptimal interventions". The *Freq-Aware Gen* strategy, by design, focuses its efforts on low-frequency groups (*e.g.*, rare classes, small objects). While this yields modest gains in those specific areas, it overlooks a larger opportunity for model improvement.

• **Observation 3: Fidelity Gap Limits Generative Data Augmentation.** While both *Bias-Agnostic Gen* and *Real Data* enrich the training set by adding new data that follows the identical biased distribution of the original 1/4 subset (*i.e.*, not attributable to the layout choices or data distribution), the mAP gain from *Real Data* is consistently higher than that of *Bias-Agnostic Gen*.

ANALYSIS: Since the data distribution is perfectly controlled, the performance gap can be directly attributed to the fidelity gap between synthesized images and real-world data. This finding supports our claim that "current L2I approaches struggle to render new samples faithfully". In this work, we will solve this problem from both the layout conditioning and generator training strategies.

> ***Remark.*** Our empirical analyses confirm that while generation-based data augmentation is promising, current approaches fall short in two aspects. **First**, the suboptimal performance of the frequency-driven *Freq-Aware Gen* strategy demonstrates that instance frequency is an incomplete proxy for the representation needs of models. A more sophisticated diagnostic tool is required to identify the true data gaps. **Second**, the performance gap between *Bias-Agnostic Gen* and *Real Data*, which both share bias of the training set, reveals a fundamental limitation in current synthesis control and fidelity. This suggests that even if we know what to generate, current layout-to-image methods lack the precision to generate it effectively.

## 3 VISUAL-PROMPTED DYNAMIC DEBIASING FOR OBJECT DETECTION

This section presents our generation-based debiasing framework, which includes a dynamic debiasing engine (§3.1) to construct unbiased layouts guided by both frequency and sample diversity, and a visual blueprint-prompted synthesis pipeline (§3.2) powered by generative alignment.

### 3.1 DYNAMIC DEBIASING VIA SCORING-DRIVEN LAYOUT GENERATION

There are two core challenges in our generation-based debiasing strategy. First, we need to quantitatively measure the dataset biases inherent, which is the foundation for targeted debiasing. Second,

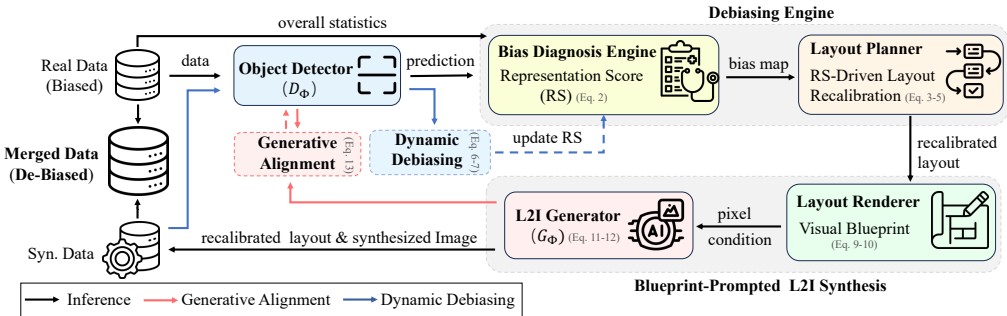

Figure 2: The overall pipeline of our framework, which 1) analyzes real data statistics to compute the representation score (Eq. 2), considering across frequency and diversity (Eq. 1); 2) performs RS-driven layout recalibration (Eq. 3-5) to sample target layouts for under-represented groups; 3) converts recalibrated layouts into visual blueprints (Eq. 9-10), which provide pixel-level conditions for L2I generation (Eq. 11-12); 4) the process is constrained by duality-aware generative alignment (Eq. 13) for feature consistency and error-based dynamic debiasing (Eq. 6-7) for adaptive RS updates.

the generated layouts for L2I synthesis should be both diverse and physically plausible, as naive randomization often produces unrealistic scenes that are unsuitable for model training.

**Representation Score.** We define a *representation score* (RS) as the quantitative proxy for how well a specific data group is represented in the dataset. Groups with low RS are under-represented and prioritized for debiasing. For object detection, the data group $\mathcal{G} = (c, s, u)$ is a set of bounding boxes with attributes including object class $c$, box size $s$, and horizontal position $u$ of box center. Considering that $s$ and $u$ are continuous variables, we discretize the image coordinates into an $M \times M$ grid for position $u$, and categorize object areas into discrete $K$ logarithmic bins for size $s$.

The sample frequency computes the empirical probability of instances in $\mathcal{G}$ occurring in an image: $\mathcal{D}_{\text{freq}}(\mathcal{G}) = N(\mathcal{G})/N_{\text{all}}$, where $N(\mathcal{G})$ is the instance number of $\mathcal{G}$ and $N_{\text{all}}$ is the number of all instances in the dataset. The analysis in §2 reveals that relying solely on instance frequency is insufficient, as even frequent groups can be underrepresented. Thus, RS moves beyond merely counting instances to integrate representation diversity, which comprises both visual and context diversity. The visual diversity $\mathcal{D}_{\text{vis}}(\mathcal{G})$ is defined as the average feature distance between instances in $\mathcal{G}$ which captures intra-group visual variation, and context diversity $\mathcal{D}_{\text{ctx}}(\mathcal{G})$ reveals the co-occurrence between class $c$ and other classes::

$$\mathcal{D}_{\text{vis}}(\mathcal{G}) = \frac{1}{|\mathcal{G}|^2} \sum_{i \in \mathcal{G}} \sum_{j \in \mathcal{G}} ||\boldsymbol{o}_i - \boldsymbol{o}_j||^2, \qquad \mathcal{D}_{\text{ctx}}(\mathcal{G}) = \frac{1}{|\mathcal{I}_{c(\mathcal{G})}| \cdot |\mathcal{C}|} \sum_{i \in \mathcal{I}_{c(\mathcal{G})}} |\mathcal{K}_i|, \qquad (1)$$

where $\boldsymbol{o}$ is extracted by the detector backbone after ROI pooling, $\mathcal{I}_{c(\mathcal{G})}$ is the set of images containing class $c$ in group $\mathcal{G}$, $\mathcal{K}_i$ is the set of classes in image $i$, and $\mathcal{C}$ is the set of all classes in the dataset. Finally, three components are combined into a representation score:

$$\text{RS}(\mathcal{G}) = \mathcal{D}_{\text{freq}}(\mathcal{G}) \cdot (\mathcal{D}_{\text{vis}}(\mathcal{G}) + \beta \cdot \mathcal{D}_{\text{ctx}}(\mathcal{G})). \qquad (2)$$

RS provides a robust measure of representation quality. Groups with low RS can then be targeted for generative debiasing, ensuring focused and effective correction of dataset imbalances.

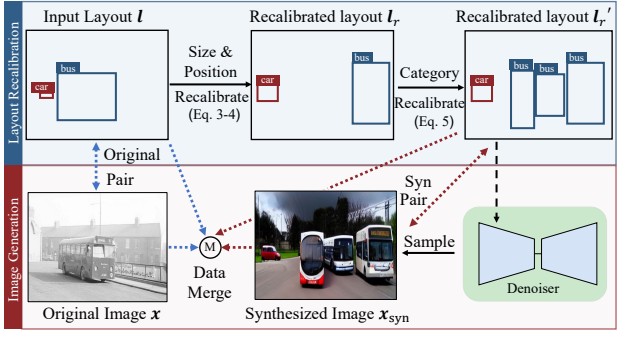

Figure 3: Illustration for layout recalibration.

**Layout Recalibration.** To preserve plausible object relations, we begin with a layout seeded from a real image, and then perturb it under the guidance of RS, to fill the identified representational gaps. Instead of recalibrating position and size independently, we treat them as coupled attributes of a data group $\mathcal{G}$ and resample them jointly. For a given object in the seed layout belonging to group $\mathcal{G} = (c, s, u)$, as shown in Fig. 3, it

is shifted to a new target group $\mathcal{G}' = (c, s', u')$ by sampling a new size $s'$ and position $u'$. The sampling probability is inversely proportional to the RS of the target group:

$$\pi(s', u' \mid c) \propto \big(\mathrm{RS}(c, s', u') + \varepsilon\big)^{-\tau}, \tag{3}$$

where $\mathrm{RS}(c, s', u')$ is the pre-computed RS for the group defined by class $c$, size bin $s'$, and position bin $u'$. The hyperparameter $\tau$ controls the strength of the debiasing. On the other hand, to preserve the natural vertical layering (*e.g.*, sky above ground, cars on roads), the vertical center $v'$ of bounding box is only slightly perturbed from its original position $v$ with a small Gaussian jitter:

$$v' = v + \epsilon, \quad \text{where} \quad \epsilon \sim \mathcal{N}(\mu = 0, \sigma^2 = (\sigma_y)^2). \tag{4}$$

$\sigma_y$ is intentionally kept small to ensure that the vertical placement of objects remains faithful to their original context. This integrated layout recalibration approach is more powerful than treating each attribute in isolation, as it respects the complex dependencies between object properties.

To enrich the dataset with underrepresented object categories, the target class $c'$ is chosen according to a context-aware, RS-guided policy that balances contextual plausibility and representation gaps:

$$\pi_c(c' \mid \mathcal{K}) \propto \underbrace{\big(\kappa \cdot \mathbf{1}[c' \in \mathcal{K}] + \mathbf{1}[c' \notin \mathcal{K}]\big)}_{\text{Context-Aware Term}} \cdot \underbrace{\big(\overline{\mathrm{RS}}(c') + \varepsilon\big)^{-\tau}}_{\text{RS-Guided Term}}, \tag{5}$$

where $\mathcal{K}$ is the set of classes already present in the scene. The context-aware term encourages adding instances of classes already present ($\kappa > 1$). $\overline{\mathrm{RS}}(c')$ is the mean representation score for class $c'$, averaged over all its size and position bins. Once a target class $c'$ is selected, we choose its specific size ($s'$) and position ($u'$) using the same inverse-RS sampling policy from Eq.3, ensuring the newly added object fills the most needed representational gap for that class.

**Error-Based Dynamic Debiasing.** The representation score (RS) provides a strong foundation for bias-aware layout recalibration, which further contributes to debiased object detection learning. However, since RS remains static throughout training, it cannot reflect the evolving bias of datasets enriched with newly generated data samples. To address this, RS should be dynamically updated to account for shifts in group-level representation qualities. Specifically, given the training procedure:

$$\boldsymbol{l}_{\text{pred}} = D_\Phi(\boldsymbol{x}_{\text{syn}}), \quad \boldsymbol{x}_{\text{syn}} = G_\Phi(\boldsymbol{l}_{\text{recalib}}), \tag{6}$$

where $\boldsymbol{l}_{\text{recalib}}$ is the layout after bias-aware recalibration, $G_\Phi$ and $D_\Phi$ represent generator and detector, respectively. The training objective of the object detector is to minimize the layout consistency loss (*i.e.*, $\mathcal{L}_{\text{layout}}$) between the predicted and the ground-truth recalibrated layouts. Crucially, the RS for each data group $\mathcal{G}_i$ is refined using an exponential moving average with $\mu = 0.99$ that incorporates the detection error $\mathcal{L}_{\text{layout}}(i)$ for instance $i$ within that group:

$$\mathrm{RS}'(\mathcal{G}_i) = \mu \cdot \mathrm{RS}(\mathcal{G}_i) + (1 - \mu) \cdot \mathcal{L}_{\text{layout}}(i). \tag{7}$$

This establishes a dynamic debiasing mechanism, where $G_\Phi$ is continuously steered to produce informative data to mitigate emerging biases, ensuring a targeted and adaptive learning process.

## 3.2 High-Fidelity L2I Synthesis with Visual Blueprints

Given a geometric layout $\boldsymbol{l} = \{(\boldsymbol{b}_n, c_n)\}_{n=1}^N \in \mathbb{R}^{N \times 5}$, composed of $N$ objects with corresponding bounding boxes $\boldsymbol{b}_n = [x_{n,1}, y_{n,1}, x_{n,2}, y_{n,2}] \in \mathbb{R}^4$ and class labels $c_n \in \mathcal{C}$, layout-to-image (L2I) synthesis (Zhao et al., 2019; Zheng et al., 2023) aims to generate visually coherent images that respect the specified structure. A common solution in existing work (Chen et al., 2024a; Wang et al., 2024) is to serialize the layout $\boldsymbol{l}$ into a token sequence $\boldsymbol{s}(\boldsymbol{l})$, which is then appended with a text prompt $y$ to form a unified conditional input $\tilde{\boldsymbol{y}} = \texttt{concat}(y, \boldsymbol{s}(\boldsymbol{l}))$. The training objective is to minimize the difference between true and predicted noise following Rombach et al. (2022):

$$\mathcal{L}_{\text{L2I}} = \mathbb{E}\left\|\boldsymbol{\epsilon} - \boldsymbol{\epsilon}_\theta\big(\boldsymbol{x}_t, \, t, \, f_\psi(\tilde{\boldsymbol{y}})\big)\right\|_2^2, \tag{8}$$

where $f_\psi$ is the text encoder. Despite being straightforward, it suffers from a textual bottleneck caused by serializing 2D spatial arrangements into a 1D text sequence. This leads to ambiguity and imprecise spatial relationships. To overcome this, we introduce **visual blueprint**, a geometry-faithful alternative using pixel-space conditioning signals for unambiguous geometric guidance.

**Blueprint Construction.** Given layout $l$, we construct a visual blueprint $\boldsymbol{I}_{\text{cond}} \in \mathbb{R}^{H \times W \times 3}$, where bounding boxes are mapped into colored rectangles indicating different instances using a rendering operator $\mathcal{R}$ (*i.e.*, Fig. 4):

$$\boldsymbol{I}_{\text{cond}} = \mathcal{R}(\boldsymbol{l}; \mathcal{P}). \tag{9}$$

Here, $\mathcal{P} = \{\mathbf{p}_i\}_{i=1}^N$ is a color palette used to differentiate object categories. To maximize the visual distinction of object classes, the colors in $\mathcal{P}$ are assigned as evenly spaced hues on the unit circle in HSV space, which are subsequently converted to RGB values via:

$$\mathbf{p}_i = \text{RGB}\big((i-1)\varphi, \ S_0, \ V_0\big), \tag{10}$$

where $\text{RGB}(H, S, V)$ is the standard HSV-to-RGB mapping, and $\varphi$ is a fixed hue step. Saturation $S$ and value $V$ are set to 1 for maximum vibrancy. However, rendering only colored rectangles can result in information loss, particularly in complex scenes containing overlapping or multiple instances of the same class. To address this, the rendering operator $\mathcal{R}$ follows three principles: **i)** to ***distinguish instances*** of the same class, the HSV value is decremented by a small step $\alpha$ for each subsequent instance; **ii)** objects are rendered in descending order of bounding-box size to prevent smaller objects from being fully ***occluded*** by larger ones; and **iii)** background objects are rendered with slight

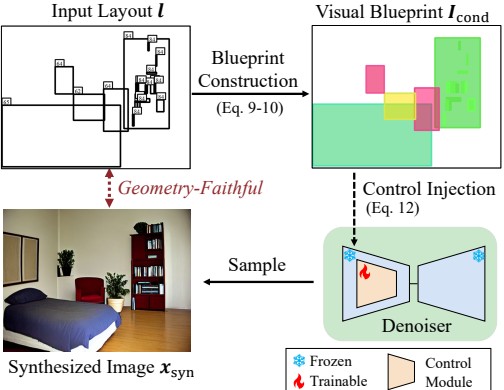

Figure 4: Illustration for blueprint construction.

transparency, so as to provide the model with visual cues about ***overlapping relationships***. It is worth noting that the binary maps used in ControlNet typically represent classes with adjacent integers. This leads to low numerical variance for different classes and introduces potential ambiguity for the encoder (*e.g.*, the Person class corresponds to (0, 0, 0) and Sheep to (0, 0, 19)). In contrast, we assign instance masks to equidistant hues on the HSV unit circle. This ensures distinct pixel values in RGB space, and provides a high-variance signal that is far easier for the encoder to distinguish. A visualization of this comparison is provided in Fig. 6 of the Appendix.

**Blueprint-Prompted Layout Conditioning.** To integrate our blueprint $\boldsymbol{I}_{\text{cond}}$ into the generation process, we require an architecture that can inject its rich spatial information into a pre-trained U-Net without sacrificing its powerful generative priors. The adapter-based strategy proposed by Zhang et al. (2023) is ideally suited for this setup. The blueprint is first projected into multi-resolution feature maps, $\boldsymbol{u} = g_\phi(\boldsymbol{I}_{\text{cond}})$, via a lightweight, trainable encoder $g_\phi$. This provides an unambiguous, multi-scale structural prior that complements the global semantic guidance from the standard text prompt $y$. The model then learns to generate the image by minimizing our visual L2I objective:

$$\mathcal{L}_{\text{visual\_L2I}} = \mathbb{E}\Big\|\boldsymbol{\epsilon} - \boldsymbol{\epsilon}_\theta\big(\boldsymbol{x}_t, \ t, \ f_\psi(y), \boldsymbol{u}\big)\Big\|_2^2. \tag{11}$$

These structural features $\boldsymbol{u}$ are then fused into the frozen U-Net $\mathcal{F}(\cdot; \Theta)$ using a trainable copy $\mathcal{F}(\cdot; \Theta_c)$, and two zero-initialized adapter blocks $\mathcal{Z}_1$ and $\mathcal{Z}_2$:

$$\mathbf{y}_c = \mathcal{F}(\boldsymbol{x}; \Theta) + \mathcal{Z}_2\left(\mathcal{F}\left(\boldsymbol{x} + \mathcal{Z}_1(\boldsymbol{u}); \Theta_c\right)\right). \tag{12}$$

As such, we treat the pre-trained diffusion model as a powerful generative backbone and specialize it for our debiasing task, guided by the unambiguous geometric information from our visual blueprint.

**Duality-Aware Generative Alignment.** Current generative frameworks treat the L2I generator and object detector as isolated components, leading to a misalignment where the synthesized image, though visually plausible, is not optimally aligned with the feature space of the detector. To bridge this gap, we propose an alignment strategy based on the duality of the two tasks.

Specifically, while the detector learns a mapping from images to layouts ($D_\Phi : \boldsymbol{x} \to \boldsymbol{l}$), the generator learns the inverse ($G_\Phi : \boldsymbol{l} \to \boldsymbol{x}$). We leverage this loop and propose an image-alignment loss $\mathcal{L}_{\text{image}}^{\text{IA}}$:

$$\mathcal{L}_{\text{image}}^{\text{IA}} = \Big\|\boldsymbol{\epsilon}_\theta\big(\boldsymbol{x}_t, \ t, \ f_\psi(y), \boldsymbol{u}\big) - \boldsymbol{\epsilon}_\theta\big(\boldsymbol{x}_t, \ t, \ f_\psi(y), \boldsymbol{u}^{\text{pred}}\big)\Big\|_2^2, \tag{13}$$

where $\boldsymbol{u}^{\text{pred}}$ is the multi-resolution feature maps constructed from the layout $\boldsymbol{l}^{\text{pred}}$ output by the detector $D_\Phi$. The final training objective for optimizing the detector is given as:

$$\mathcal{L}_{\text{OD}} = \mathcal{L}_{\text{det}} + \lambda \mathcal{L}_{\text{image}}^{\text{IA}}, \tag{14}$$

where $\mathcal{L}_{\text{det}}$ is the conventional object detection loss, $\lambda$ is a balance factor. As such, $\mathcal{L}_{\text{image}}^{\text{IA}}$ penalizes the detector for producing layouts that are insufficient for faithful image synthesis, and forces the detector to be robust to the features produced by the generator to deliver consistent predictions

## 4 RELATED WORK

**Dataset Biases and Debiasing.** Dataset bias occurs when training data is not representative samples of the real-world scenarios. This misalignment causes models to learn dataset-specific shortcuts instead of generalizable features (Torralba & Efros, 2011; Geirhos et al., 2020). Efforts to mitigate dataset bias largely fall into two categories. Data-based strategies resample or re-weight the training distribution to give more importance to rare instances (Cui et al., 2019; Cao et al., 2019). In contrast, learning-based strategies dynamically adjust gradients to prevent common classes from dominating the learning process (Tan et al., 2020; Wang et al., 2021). In object detection, these biases manifest across axes like long-tailed category distributions where a few classes dominate the dataset (Ouyang et al., 2016), object size skew that favors normal and large instances over small ones (Herranz et al., 2016; Gilg et al., 2023), and spatial bias where objects concentrate in center image zones (Zheng et al., 2024). Accordingly, solutions commonly use resampling and re-balancing to enhance rare categories (Gupta et al., 2019; Tan et al., 2021), scale-aware architectures to boost small objects (Lin et al., 2017; Singh & Davis, 2018; Singh et al., 2018), or copy-paste to increase the sample quantities (Ghiasi et al., 2021). Despite the success, these approaches are primarily frequency-centric, treating instance counts as the main proxy for biases. In this work, we propose a generation-based debiasing strategy, which contains a new image synthesis architecture, a bias-aware layout sampling strategy, and a dynamic engine that adapts to evolving biases during training.

**Controllable Diffusion Models.** Diffusion probabilistic models (Sohl-Dickstein et al., 2015) have developed rapidly in recent years (Dhariwal & Nichol, 2021; Ho & Salimans, 2022; Kingma et al., 2021; Rombach et al., 2022). Owing to their exceptional generation quality and controllability, diffusion models now become the dominant paradigm across a range of applications, including image editing (Brooks et al., 2023; Kawar et al., 2023; Meng et al., 2021; Hertz et al., 2022), image-to-image translation (Saharia et al., 2022a; Tumanyan et al., 2023; Li et al., 2023a), and text-to-image (T2I) generation (Nichol et al., 2021; Podell et al., 2023; Rombach et al., 2022; Saharia et al., 2022b; Li et al., 2024a; Jin et al., 2025), *etc*. Recent layout-to-image (L2I) synthesis (Zhao et al., 2019; Li et al., 2021; Yang et al., 2022; Sun & Wu, 2019) aims at precise, instance-level placement by augmenting pre-trained T2I models with layout information (*i.e.*, bounding boxes and category labels). Specifically, the layout is converted into a text token sequence and then injected into a pre-trained T2I diffusion model (Cheng et al., 2023; Yang et al., 2023; Couairon et al., 2023; Xie et al., 2023; Chen et al., 2024b; Wang et al., 2025; Li et al., 2025; Cai et al., 2025). While this approach offers scalability, it introduces a textual bottleneck in which 2D spatial arrangements are converted into 1D text sequences. Departing from this paradigm, our method encodes layouts in pixel-space as visual blueprint images. This provides the model with direct and unambiguous spatial and relational instructions to guide the generation process with high fidelity and controllability.

**Generation-Based Data Augmentation.** Advanced strategies seek to enhance model generalization beyond simple resampling. Mixing-based techniques regularize model training by virtual samples created from interpolated images and labels (Zhang et al., 2018) or substituted regional patches (Yun et al., 2019). Erasure-based methods improve robustness by randomly masking image regions (De-Vries & Taylor, 2017; Zhong et al., 2020). While label-preserving and simple to deploy, these methods only recombine visual patterns already present in the training data, thereby constraining the diversity of generated samples. In contrast, recent work (Zhao et al., 2023; Suri et al., 2023; Chen et al., 2024a; Li et al., 2024b; Xiang et al., 2024) explores using synthetic data from generative models to improve model performance. For example, X-Paste (Zhao et al., 2023) scales copy-paste by synthesizing instances with diffusion models. Gen2Det (Suri et al., 2023) leverages conditioned diffusion to directly synthesize scene-specific images. Layout-to-image synthesis (Chen et al., 2024a; Wang et al., 2024) reuses layouts in the training set and applies flip augmentation to synthesize additional samples for the detector. In contrast to these bias-agnostic approaches, this work introduces

Table 1: Quantitative results for fidelity on MS COCO (Lin et al., 2014) and NuImages (Caesar et al., 2020).

| Model | Res. | MS COCO | | | | NuImages | | | | | |
|---|---|---|---|---|---|---|---|---|---|---|---|
| | | FID ↓ | mAP ↑ | $AP_{50}$ ↑ | $AP_{75}$ ↑ | FID ↓ | mAP ↑ | $AP_{50}$ ↑ | $AP_{75}$ ↑ | $AP^m$ ↑ | $AP^l$ ↑ |
| Real Image | - | - | 48.9 | 68.3 | 55.6 | - | 48.2 | 75.0 | 52.0 | 46.7 | 60.5 |
| LAMA (Li et al., 2021) | $256^2$ | 31.12 | 13.4 | 19.7 | 14.9 | 63.85 | 3.2 | 8.3 | 1.9 | 2.0 | 9.4 |
| Taming (Jahn et al., 2021) | | 33.68 | - | - | - | 32.84 | 7.4 | 19.0 | 4.8 | 2.8 | 18.8 |
| TwFA (Yang et al., 2022) | | 22.15 | - | 28.2 | 20.1 | - | - | - | - | - | - |
| GeoDiffusion (Chen et al., 2024a) | | 20.16 | 29.1 | 38.9 | 33.6 | 14.58 | 15.6 | 31.7 | 13.4 | 6.3 | 38.3 |
| DetDiffusion (Wang et al., 2024) | | 19.28 | 29.8 | 38.6 | 34.1 | - | - | - | - | - | - |
| GDCC (Cai et al., 2025) | | 18.02 | 31.4 | 41.2 | 36.4 | 12.54 | 17.5 | 33.5 | 15.6 | 8.3 | 40.3 |
| **Ours** | | **16.35** | **33.6** | **46.6** | **36.8** | **12.43** | **19.8** | **38.9** | **16.9** | **10.8** | **43.2** |
| ReCo (Yang et al., 2023) | $512^2$ | 29.69 | 18.8 | 33.5 | 19.7 | 27.10 | 17.1 | 41.1 | 11.8 | 10.9 | 36.2 |
| GLIGEN (Li et al., 2023b) | | 21.04 | 22.4 | 36.5 | 24.1 | 16.68 | 21.3 | 42.1 | 19.1 | 15.9 | 40.8 |
| ControlNet (Zhang et al., 2023) | | 28.14 | 25.2 | 46.7 | 22.7 | 23.26 | 22.6 | 43.9 | 20.7 | 17.3 | 41.9 |
| GeoDiffusion (Chen et al., 2024a) | | 18.89 | 30.6 | 41.7 | 35.6 | 9.58 | 31.8 | 62.9 | 28.7 | 27.0 | 53.8 |
| GDCC (Cai et al., 2025) | | 17.15 | 32.6 | 43.6 | 38.0 | 7.97 | 33.6 | 64.7 | 30.7 | 28.6 | 55.9 |
| **Ours** | | **15.24** | **46.5** | **61.4** | **51.6** | **8.35** | **40.2** | **70.1** | **38.2** | **38.4** | **58.0** |

Table 2: Quantitative results for debiasing on MS COCO (Lin et al., 2014) *w.r.t.* different attributes.

| Model | mAP ↑ | center ↑ | middle ↑ | outer ↑ | freq ↑ | comm ↑ | rare ↑ | large ↑ | normal ↑ | small ↑ |
|---|---|---|---|---|---|---|---|---|---|---|
| Faster R-CNN (Baseline) | 37.4 | 37.7 | 33.9 | 28.3 | 31.2 | 38.9 | 43.2 | 48.1 | 41.0 | 21.2 |
| *Bias Agnostic* | | | | | | | | | | |
| Copy Paste (Ghiasi et al., 2021) | 37.9 | 38.2 | 35.5 | 28.8 | 31.4 | 39.4 | 43.6 | 48.8 | 41.5 | 21.5 |
| ControlNet (Zhang et al., 2023) | 36.9 | 37.3 | 33.4 | 27.6 | 30.8 | 38.3 | 42.9 | 49.0 | 40.4 | 19.8 |
| GeoDiffusion (Chen et al., 2024a) | 38.4 | 38.6 | 35.0 | 29.5 | 32.0 | 39.9 | 44.3 | 50.3 | 42.1 | 19.7 |
| *Frequency Aware* | | | | | | | | | | |
| ControlNet + Resampling | 36.9 ↓0.5 | 37.2 ↓0.5 | 33.4 ↓0.5 | 27.9 ↓0.4 | 30.2 ↓1.0 | 37.7 ↓0.8 | 43.2 ↓0.0 | 48.6 ↑0.5 | 40.5 ↓0.5 | 20.1 ↓1.1 |
| GeoDiffusion + Resampling | 38.5 ↑1.1 | 38.5 ↑0.8 | 35.3 ↑1.4 | 30.0 ↑1.7 | 31.6 ↑0.4 | 39.4 ↑0.5 | 44.5 ↑1.3 | 49.9 ↑1.8 | 42.2 ↑1.2 | 20.0 ↓1.2 |
| **Ours** | **40.3** ↑2.9 | **40.5** ↑2.8 | **36.9** ↑3.0 | **31.5** ↑3.2 | **33.3** ↑2.1 | **41.8** ↑2.9 | **46.8** ↑3.6 | **52.5** ↑4.4 | **43.8** ↑2.8 | **23.1** ↑1.9 |

a bias-aware data augmentation framework. We begin by systematically diagnosing dataset biases across key axes including spatial location, category frequency, and object size. Inspired by this analysis, we design a bias-aware layout sampling strategy, ensuring that the generated data is not only diverse but also precisely aligned with the goal of mitigating specific, pre-identified dataset biases.

## 5 EXPERIMENT

**Experimental Setup.** Following existing work (Chen et al., 2024a; Wang et al., 2024), the validation contains two setups: **Fidelity**: which assesses the quality of generated images by applying pretrained detection models to images synthesized from ground-truth layouts in the validation set, using the proposed L2I model. We report the Fréchet Inception Distance (FID) to assess generation quality and mean Average Precision (mAP) to measure detection performance. **Debiasing**: which evaluates the ability of generated data to mitigate biased distributions across data groups. The baselines are SOTA L2I models, which synthesize new training sets using annotations from real training samples, with layout augmentations limited to random flip and slight perturbation (*i.e.*, *bias-agnostic*). On this basis, we construct frequency-aware variants by relaxing the layout augmentations to include the resampling strategy Gupta et al. (2019) (*i.e.*, *frequency-aware*). Finally, we compare them against our proposed dynamic-debiasing and visual prompted L2I synthesis approach. To evaluate debiasing effectiveness, we measure not only the overall mAP but also the performance across spatial positions (*i.e.*, $mAP_{\{center,middle,outer\}}$), category frequency (*i.e.*, $mAP_{\{frequent,common,rare\}}$), and object size (*i.e.*, $mAP_{\{large,normal,small\}}$). For all experiments, unless otherwise specified, we employ the Faster R-CNN (Ren et al., 2015) with a ResNet-50 backbone (He et al., 2016). More implementation details regarding network architecture, training, testing, and training objectives are provided in *Appendix*.

**Dataset.** Our proposed L2I synthesis model and corresponding debiasing strategy are evaluated on **MS COCO** (Lin et al., 2014) which provides 118K training and 5K validation images for over 80 object categories, and **NuImages** (Caesar et al., 2020) which is derived from the nuScenes autonomous driving benchmark, containing 60K training and 15K validation samples from 10 semantic classes.

### 5.1 EXPERIMENTAL RESULTS

**Fidelity.** Our approach achieves significantly higher performance in fidelity (Table 1), surpassing prior SOTA (*i.e.*, GeoDiffusion (Chen et al., 2024a)) by **15.9** mAP, **19.7** $AP_{50}$, **16.0** $AP_{75}$ on MS

Table 3: Quantitative results for debiasing on NuImages (Caesar et al., 2020) *w.r.t.* low-performing categories.

| Model | mAP ↑ | outer ↑ | rare ↑ | large ↑ | small ↑ | trailer ↑ | const. ↑ | ped. ↑ | cone ↑ |
|---|---|---|---|---|---|---|---|---|---|
| Faster R-CNN (Baseline) | 36.9 | 27.9 | 38.5 | 50.7 | 25.1 | 15.5 | 24.0 | 31.3 | 32.5 |
| *Bias Agnostic* | | | | | | | | | |
| Copy Paste (Ghiasi et al., 2021) | 37.5 | 28.6 | 38.8 | 51.5 | 25.3 | 16.0 | 24.7 | 31.5 | 32.7 |
| ControlNet (Zhang et al., 2023) | 36.4 | 27.6 | 38.3 | 51.2 | 24.4 | 13.6 | 24.1 | 30.3 | 31.8 |
| GeoDiffusion (Chen et al., 2024a) | 38.3 | 28.4 | 39.6 | 52.4 | 25.3 | 18.3 | 27.6 | 30.5 | 32.1 |
| *Frequency Aware* | | | | | | | | | |
| ControlNet + Resampling | 36.5 ↓0.4 | 27.9 ↓0.4 | 38.5 ↓0.4 | 51.0 ↑0.3 | 24.5 ↓0.4 | 13.6 ↓0.4 | 24.2 ↓0.4 | 30.4 ↓0.4 | 31.9 ↓0.4 |
| GeoDiffusion + Resampling | 38.3 ↑1.4 | 28.8 ↑0.9 | 40.0 ↑0.5 | 52.0 ↑1.3 | 25.4 ↑0.3 | 18.0 ↑2.5 | 27.5 ↑3.5 | 30.8 ↓0.5 | 32.3 ↓0.8 |
| **Ours** | **40.0** ↑3.1 | **31.5** ↑3.6 | **42.5** ↑4.0 | **54.8** ↑4.1 | **27.4** ↑2.3 | **19.5** ↑4.0 | **29.7** ↑5.7 | **32.1** ↑0.8 | **33.0** ↑0.5 |

Figure 5: Visualization of recalibrated layouts, showing objects with updated sizes and positions, and new instances (left). Our method can generate geometry-faithful images compared to prior SOTA (right).

COCO, and **8.4** mAP, **11.4** $AP^m$, **4.2** $AP^l$ on NuImages, under the $512^2$ resolution. It also yields much lower FID scores (*i.e.*, **15.24** *vs.* 18.89 of GeoDiffusion on MS COCO), verifying the effectiveness of our blueprint-prompted synthesis and generative alignment strategies.

**Debiasing.** As seen in Tables 2-3, bias-agnostic methods including copy-paste (Ghiasi et al., 2021), ControlNet (Zhang et al., 2023), and GeoDiffusion (Chen et al., 2024a), boost performance broadly but are ineffective for underrepresented groups, leading to a modest enhancement in the final mAP. Meanwhile, integrating generative methods with the resampling strategy (Gupta et al., 2019) offers certain improvement for underrepresented groups. Our approach, by targeting biases through both frequency and representation diversity, delivers substantial improvements across the board. It not only achieves significant performance gains for underrepresented groups (*e.g.*, 28.3→31.5 for $mAP_{outer}$, 43.2→46.8 for $mAP_{rare}$ on MS COCO), but also sets new SOTAs for overall scores, achieving 40.3 and 40.0 mAP on MS COCO and NuImages, respectively. The comprehensive results validate the overall design and confirm the powerful debiasing effectiveness of our method.

**Qualitative Results.** As shown in Fig. 5, our method can adjust object sizes and locations, and even add new instances according to model needs. Moreover, it can generate geometry-faithful images with complicated layouts containing over ten instances, outperforming prior SOTA.

## 5.2 DIAGNOSTIC EXPERIMENTS

We conduct a series of ablation studies on MS COCO, all under the **Debiasing** setup.

**Essential Components.** We examine the efficacy of essential components in Table 4. After replacing textual layout conditions with visual blueprints, the mAP enjoys large improvement (37.0→ 38.9), indicating the effective preservation of spatial cues. Generative alignment enjoys moderate improvements, as its primary role is to enhance the fidelity of generated images, rather than directly boosting detection performance. Meanwhile, RS-driven layout recalibration and dynamic debiasing also deliver satisfactory improvements, particularly benefiting underrepresented data groups.

**Dynamic Debiasing.** We ablate the momentum parameter $\mu$ for dynamic debiasing in Table 9. A value of 0, which updates RS using only errors from the current batch, leads to unstable training and poor performance. Conversely, $\mu = 1$ disables the dynamic update and reverts to a static RS. We found that $\mu = 0.99$ achieves the best performance. This demonstrates a stable yet responsive update for RS to dynamically reflect the evolving representation quality and mitigate emerging biases.

**Conditional Input.** We evaluate the impact of layout conditions on both synthesis fidelity (Table 6) and debiasing performance (Table ref). As shown in Table 6, replacing textual inputs with Visual

Table 4: Ablative studies of essential components in our proposed method on MS COCO 2017 (Lin et al., 2014).

| Method | mAP↑ | outer ↑ | rare ↑ | large ↑ | small↑ |
|---|---|---|---|---|---|
| Baseline | 37.0 | 27.8 | 43.0 | 47.9 | 20.5 |
| + Visual Blueprint | 38.9 | 29.6 | 45.0 | 51.1 | 21.9 |
| + Generative Align. | 39.1 | 29.9 | 45.2 | 51.3 | 22.1 |
| + RS-Driven Recali. | 39.9 | 31.0 | 46.4 | 52.3 | 22.8 |
| + Dynamic Debias. | 40.3 | 31.5 | 46.8 | 52.5 | 23.1 |

Table 5: Ablative studies of dynamic debiasing on MS COCO 2017 (Lin et al., 2014).

| $\mu$ | mAP↑ | outer ↑ | rare ↑ | large ↑ | small↑ |
|---|---|---|---|---|---|
| 0 | 38.6 | 29.4 | 44.2 | 49.7 | 21.5 |
| 0.9 | 40.0 | 31.1 | 46.2 | 51.9 | 23.0 |
| 0.99 | 40.3 | 31.5 | 46.8 | 52.5 | 23.1 |
| 0.999 | 40.1 | 31.3 | 46.4 | 52.0 | 22.8 |
| 1 | 39.8 | 31.0 | 46.4 | 51.6 | 22.5 |

Table 6: Ablative studies of Blueprint design for L2I synthesis on MS COCO 2017 (Lin et al., 2014).

| Method | FID↓ | mAP ↑ | $AP_{50}$ ↑ | $AP_{75}$ ↑ |
|---|---|---|---|---|
| Baseline | 28.14 | 25.2 | 46.7 | 22.7 |
| + Pixel Canvas | 20.15 | 40.8 | 56.2 | 40.5 |
| + Instance Discrim. | 17.05 | 44.5 | 59.5 | 48.8 |
| + Overlap Aware. | 15.24 | 46.5 | 61.4 | 51.6 |

Table 7: Ablative studies of Blueprint design for debiasing on MS COCO 2017 (Lin et al., 2014).

| Method | mAP↑ | outer ↑ | rare ↑ | large ↑ | small↑ |
|---|---|---|---|---|---|
| Textual Layout | 37.0 | 27.8 | 43.0 | 47.9 | 20.5 |
| Pixel Canvas | 38.5 | 29.1 | 44.6 | 50.5 | 21.4 |
| + Instance Discrim. | 38.7 | 29.4 | 44.8 | 50.7 | 21.7 |
| + Overlap Aware. | 38.9 | 29.6 | 45.0 | 51.1 | 21.9 |

Table 8: Ablative studies of representation score for layout generation on MS COCO 2017 (Lin et al., 2014).

| Score | mAP↑ | outer ↑ | rare ↑ | large ↑ | small↑ |
|---|---|---|---|---|---|
| Bias-Agnostic | 39.1 | 29.9 | 45.2 | 51.3 | 22.1 |
| $\mathcal{D}_{\text{freq}}$ | 39.3 | 30.4 | 45.8 | 50.9 | 22.5 |
| $\mathcal{D}_{\text{freq}} + \mathcal{D}_{\text{vis}}$ | 39.7 | 30.7 | 46.3 | 52.0 | 22.6 |
| $\mathcal{D}_{\text{freq}} + \mathcal{D}_{\text{ctx}}$ | 39.5 | 30.9 | 46.1 | 51.2 | 22.7 |
| $\mathcal{D}_{\text{vis}} + \mathcal{D}_{\text{ctx}}$ | 39.5 | 30.6 | 45.9 | 51.7 | 22.4 |
| $\mathcal{D}_{\text{freq}} + \mathcal{D}_{\text{vis}} + \mathcal{D}_{\text{ctx}}$ | 39.9 | 31.0 | 46.4 | 52.3 | 22.8 |

Table 9: Experiments of more detectors on MS COCO 2017 (Lin et al., 2014).

| Detector | mAP↑ | outer ↑ | rare ↑ | large ↑ | small↑ |
|---|---|---|---|---|---|
| YOLOX-s | 40.5 | 31.2 | 44.5 | 53.1 | 23.5 |
| + Ours | 43.3 ↑2.8 | 34.5 | 48.1 | 56.2 | 25.1 |
| DINO | 49.0 | 40.5 | 50.5 | 64.0 | 31.4 |
| + Ours | 51.8 ↑2.8 | 44.1 | 54.2 | 67.2 | 33.0 |
| CO-DETR | 52.0 | 43.5 | 53.5 | 67.1 | 34.8 |
| + Ours | 54.6 ↑2.6 | 46.8 | 57.2 | 70.0 | 36.3 |

Blueprints significantly improves generation quality (FID decreases from 28.14 to 20.15). Adding instance discrimination and occlusion awareness further enhances fidelity. Crucially, Table 7 shows that this improved generation quality directly translates to better detection performance (mAP increases from 37.0 to 38.9), validating our blueprint design.

**Representation Score.** We probe the design of representation score (RS) in Table 8. Relying solely on sample frequency ($\mathcal{D}_{\text{freq}}$) yields limited gains. Incorporating visual ($\mathcal{D}_{\text{vis}}$) or contextual diversity ($\mathcal{D}_{\text{ctx}}$) alongside frequency notably improves performance (*e.g.*, $\mathcal{D}_{\text{vis}} + \mathcal{D}_{\text{ctx}}$ improves mAP from 39.3 to 39.7). The best performance is achieved with the full RS ($\mathcal{D}_{\text{freq}} + \mathcal{D}_{\text{vis}} + \mathcal{D}_{\text{ctx}}$), demonstrating that a comprehensive scoring of both frequency and diversity is essential for effective debiasing.

**Generalizability.** To verify the general effectiveness of our framework beyond Faster R-CNN, We extend the evaluation to diverse modern detection paradigms, including YOLOX Ge et al. (2021), DINO Zhang et al. (2022), and CO-DETR Zong et al. (2023). As seen in Table 9, our method consistently yields performance gains of approximately 2.6 ∼ 2.8 mAP across all architectures.

**Component Dependence.** We investigate the interdependence between RS-driven debiasing (**what to generate**) and visual blueprint (**how to generate** faithfully) in Table 10. As seen, pairing our debiasing engine with visual blueprint unlocks higher performance boosts (+1.4 mAP) than pairing it with prior textprompted L2I generators (*e.g.*, GeoDiffusion). This confirms that high-fidelity synthesis is the prerequisite for unlocking the benefits of targeted debiasing.

Table 10: Analysis on the interdependence between debiasing and generation strategies.

| Method | Generator | mAP↑ |
|---|---|---|
| Baseline | - | 37.4 |
| GeoDiffusion | Text | 38.4 |
| + Scoring-Driven Debiasing | Text | 38.7 |
| Visual Blueprint (Ours) | Visual | 38.9 |
| + Scoring-Driven Debiasing | Visual | **40.3** |

## 6 CONCLUSION

In this work, we demonstrate that instance frequency is an incomplete proxy for representation needs and existing L2I synthesis methods suffer from a fidelity gap. To overcome these challenges, we proposed a scoring-driven debiasing engine, which captures both sample density and diversity to recalibrate layouts for sample generation. Furthermore, we replace ambiguous text prompts with visual blueprints and integrate a duality-aware, generative alignment strategy. This contributes to high-fidelity and geometry-faithful synthesis of targeted samples. Empirical results reveal a significant improvement in object detection performance and a reduction in bias across data groups.

**Acknowledgement.** This work was supported by Zhejiang Provincial Natural Science Foundation of China (No. LD25F020001), Fundamental Research Funds for the Central Universities (226-2025-00057), Beijing Natural Science Foundation (L252036), the National Natural Science Foundation of China (No. 62506169, 62472222), Natural Science Foundation of Jiangsu Province (No. BK20240080), National Defense Science and Technology Industry Bureau Technology Infrastructure Project (JSZL2024606C001), the Open Project Program of State Key Laboratory of Virtual Reality Technology and Systems, Beihang University (No.VRLAB2025A02), and Jiangsu Provincial Scientific Research Center of Applied Mathematics (No. BK20233002).

**Ethics statement.** Our research utilizes publicly available datasets (MS COCO and NuImages) and does not involve human subject studies or sensitive data. While image generative models can be misused, our framework is designed for data augmentation. It aims to mitigate biases in widely used technologies such as object detection, rather than to enable unrestricted image generation.

**Reproducibility.** The implementation details, including the network architecture, training objective, training and testing strategies, are provided in *Appendix*. Moreover, to ensure the reproducibility of our proposed approach, the implementation code is publicly available at `https://github.com/NUST-Machine-Intelligence-Laboratory/Beyond_Freq`.

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

# A APPENDIX

## A.1 USE OF LARGE LANGUAGE MODELS (LLMs)

We confirm that LLMs were used solely for minor grammatical corrections and phrasing suggestions. They were not involved in providing research ideas, including motivation, algorithm design, or the development of the core method. Furthermore, they were not used in generating any scientific content, such as the introduction, methodology, or experimental results presented in this paper.

## A.2 METRIC DEFINITION

For spatial location, we partition images into center, middle, and outer regions, each covering 33% of the image area, and then compute mAP for bounding boxes whose centers fall within corresponding regions, yielding $\text{mAP}_{\text{center}}$, $\text{mAP}_{\text{middle}}$, and $\text{mAP}_{\text{outer}}$. For object category, we group the 30% most occurring categories as *frequent*, 30% least occurring categories as *rare*, and the remaining as *common*, yielding $\text{mAP}_{\text{frequent}}$, $\text{mAP}_{\text{rare}}$, and $\text{mAP}_{\text{common}}$. For object size, we group the objects with the size of bounding box larger than 96×96 as *large*, smaller than 32×32 as *small*, and the remaining as *normal*, yielding $\text{mAP}_{\text{large}}$, $\text{mAP}_{\text{small}}$, and $\text{mAP}_{\text{normal}}$.

## A.3 EXPERIMENTAL SETUP

**Training.** For all detection experiments, unless otherwise specified, we employ the Faster R-CNN (Ren et al., 2015) with a ResNet-50 backbone (He et al., 2016). The models are trained following the standard $1\times$ schedule using a batch size of 16 and an initial learning rate of 0.02. For debiasing experiments, we merge the debiasing datasets with the original training sets into a unified training set. The L2I synthesis model is built upon Stable Diffusion (Rombach et al., 2022), pre-trained on LAION-5B (Schuhmann et al., 2022). The model is first trained for 100,000 iterations on 256×256 resolution images. The resulting checkpoint is then used to initialize the 512×512 model, which is subsequently fine-tuned. Both resolutions use a batch size of 16 and a constant learning rate of 1e-5.

**Testing.** To assess generation fidelity, we adhere to the protocol established in prior work (Li et al., 2021; Chen et al., 2024a). For MS COCO, we filter the validation set to include only images containing 3 to 8 objects, resulting in a split of 3,097 images, which are then evaluated using a pre-trained YOLOv4 detector (Bochkovskiy et al., 2020). For NuImages, the validation set is filtered to images with no more than 22 objects, yielding a total of 14,772 images, which are evaluated using a Mask R-CNN (He et al., 2017). Test-time augmentation is disabled for all evaluations.

**Training Objective.** For L2I synthesis models, we optimize it with the $\mathcal{L}_{\text{visual\_L2I}}$ defined in Eq. 11, while for object detection, we optimize the detector with $\mathcal{L}_{\text{OD}}$ defined in Eq. 14.

**Quantization of Representation Score.** For box size $s$, we discretize object sizes into small, medium, and large three categories following the standard MS COCO definitions (Small: area $< 32^2$; Medium: $32^2 \leq \text{area} \leq 96^2$; Large: area $> 96^2$). For horizontal position $u$, we normalize the horizontal center coordinate to $[0, 1]$, and discretize it into $K = 10$ uniform bins, to ensure each group $\mathcal{G}$ contains statistically significant samples for RS calculation.

**Synthesized Debiasing Dataset.** To facilitate a fair comparison with prior L2I synthesis methods, the scale of generated debiasing samples is aligned with the original MS COCO and NuImages training sets, comprising 120K/60K images and 840K/540K instances, respectively.

## A.4 ADDITIONAL ANALYSIS

**Ablation on Recalibration Strategy.** We examine the effectiveness of bias-aware layout recalibration in Table S1. A bias-agnostic strategy, which randomly recalibrates layouts, yields modest improvements across metrics. In contrast, targeting biases along a single attribute leads to a large improvement for its corresponding metric but only modest gains for others. Our full strategy, which jointly considers all attributes for layout recalibration, achieves the best overall performance.

Table S1: Ablative studies of recalibration strategy for layout generation on MS COCO 2017 (Lin et al., 2014).

| Attribute | mAP ↑ | outer ↑ | rare ↑ | large ↑ | small ↑ |
|---|---|---|---|---|---|
| Bias-Agnostic | 39.1 | 29.9 | 45.2 | 50.4 | 22.3 |
| Position | 39.4 | 30.9 | 45.6 | 50.6 | 22.7 |
| Size | 39.6 | 30.0 | 45.6 | 51.8 | 21.9 |
| Category | 39.5 | 30.1 | 46.3 | 50.6 | 22.6 |
| All | 39.9 | 31.0 | 46.4 | 52.3 | 22.8 |

**Ablation on Hyperparameter $\beta$.** We further investigate the impact of the hyperparameter $\beta$ in Eq. 2, which balances the weight between visual diversity and context diversity. As shown in Table S3, when $\beta = 0$, the context diversity term $D_{\text{ctx}}$ is disabled. The performance remains stable as $\beta$ varies, achieving optimal results at $\beta = 1.0$, which is adopted as our default setting.

Table S2: Ablative study of hyperparameter $\beta$ in Eq. 2.

| $\beta$ Value | 0.0 | 0.5 | 1.0† | 1.5 | 2.0 |
|---|---|---|---|---|---|
| $\text{AP}_{\text{outer}}$ | 30.4 | 30.7 | **30.9** | 30.8 | 30.5 |
| mAP | 39.7 | 39.8 | **39.9** | 39.9 | 39.7 |

Table S3: Stability analysis over 3 independent runs.

| Method | Mean mAP↑ | Std Dev↓ |
|---|---|---|
| w/o Alignment | 40.1 | 0.32 |
| w/ Alignment | **40.3** | **0.06** |

**Analysis of Generative Alignment.** We further analyze the impact of duality-aware generative alignment on training stability. Synthetic data often introduces stochastic artifacts that vary with random seeds. Without alignment constraints, the detector risks overfitting to these artifacts, leading to high variance across runs. The alignment loss ($\mathcal{L}_{\text{image}}^{\text{IA}}$) acts as a structural anchor to enforce feature consistency. To verify this, we conducted three independent training runs with different random seeds. As shown in Table S4, the inclusion of generative alignment reduces the standard deviation (from 0.32 to 0.06), which demonstrates improved robustness against generative noise.

**Debiasing.** In Table S4, we present a comparison of our proposed method against more bias-agnostic generative data augmentation approaches. As shown, our method outperforms prior work across all metrics, further demonstrating the effectiveness of our design.

## A.5 VISUAL BLUEPRINT.

Standard simple class maps typically represent categories using adjacent integer IDs, leading to low numerical variance. For example, the *Person* class corresponds to RGB values $(0, 0, 0)$ and *Sheep* to $(0, 0, 19)$. As shown in the left panel, these numerically proximate values appear visually indistinguishable (near-black), providing a weak signal that may cause ambiguity for the encoder. In contrast, our Visual Blueprint projects class labels onto the HSV unit circle to maximize signal separation. This results in significantly distinct pixel values: *Person* is rendered as $(255, 19, 0)$ and *Sheep* as $(255, 215, 0)$. As shown in the Fig. 6, this approach creates a high-variance, unambiguous signal that is significantly easier for the encoder to discriminate.

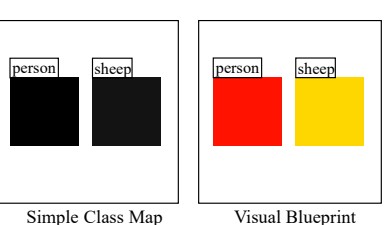

Simple Class Map          Visual Blueprint

Figure 6: Comparison of layout representations. Left: Simple class map with low numerical variance. Right: Our Visual Blueprint with high-variance RGB signals for better discrimination.

## A.6 COMPUTATIONAL COST ANALYSIS

The computational cost of our framework consists of three components: (i) **debiasing engine**, which computes representation scores involves only basic statistical operations on low-dimensional layout data, introducing negligible overhead to the pipeline; (ii) **blueprint-prompted L2I synthesis**, which involves the Layout Renderer and the L2I Generator. The Layout Renderer is computationally lightweight, while the L2I Generator (fine-tuned via adapters) requires training time comparable to ControlNet (Zhang et al., 2023). In our implementation, training the generator on MS COCO (512×512) takes approximately 74 GPU hours. In contrast, prior generation-based augmentation SOTA like GeoDiffusion needs 640 GPU hours for training.

Table S4: Quantitative results for debiasing on MS COCO (Lin et al., 2014) and NuImages (Caesar et al., 2020) with more bias-agnostic L2I synthesis methods.

| Model | MS COCO | | | | | NuImages | | | | | |
|---|---|---|---|---|---|---|---|---|---|---|---|
| | mAP ↑ | $AP_{50}$ ↑ | $AP_{75}$ ↑ | $AP^m$ ↑ | $AP^l$ ↑ | mAP ↑ | car ↑ | truck ↑ | bus ↑ | ped. ↑ | cone↑ |
| Faster R-CNN (Baseline) | 37.4 | 58.1 | 40.4 | 41.0 | 48.1 | 36.9 | 52.9 | 40.9 | 42.1 | 31.3 | 32.5 |
| *Bias Agnostic* | | | | | | | | | | | |
| LostGAN (Sun & Wu, 2019) | - | - | - | - | - | 35.6 | 51.7 | 39.6 | 41.3 | 30.0 | 31.6 |
| LAMA (Li et al., 2021) | - | - | - | - | - | 35.6 | 51.7 | 39.2 | 40.5 | 30.0 | 31.3 |
| Taming (Jahn et al., 2021) | - | - | - | - | - | 35.8 | 51.9 | 39.3 | 41.1 | 30.4 | 31.6 |
| ReCo (Yang et al., 2023) | - | - | - | - | - | 36.1 | 52.2 | 40.9 | 41.8 | 29.5 | 31.2 |
| L.Diffusion (Zheng et al., 2023) | 36.5 | 57.0 | 39.5 | 39.7 | 47.5 | - | - | - | - | - | - |
| L.Diffuse (Cheng et al., 2023) | 36.6 | 57.4 | 39.5 | 40.0 | 47.4 | - | - | - | - | - | |
| GLIGEN (Li et al., 2023b) | 36.8 | 57.6 | 39.9 | 40.3 | 47.9 | 36.3 | 52.8 | 40.7 | 42.0 | 30.2 | 31.7 |
| ControlNet (Zhang et al., 2023) | 36.9 | 57.8 | 39.6 | 40.4 | 49.0 | 36.4 | 52.8 | 40.5 | 42.1 | 30.3 | 31.8 |
| GeoDiffusion (Chen et al., 2024a) | 38.4 | 58.5 | 42.4 | 42.1 | 50.3 | 38.3 | 53.2 | 43.8 | 45.0 | 30.5 | 32.1 |
| *Frequency Aware* | | | | | | | | | | | |
| ControlNet + Resampling | 36.9 | 57.8 | 39.7 | 40.5 | 47.6 | 36.5 | 53.1 | 40.3 | 41.6 | 30.4 | 31.9 |
| GeoDiffusion + Resampling | 38.5 | 58.6 | 42.4 | 42.2 | 49.9 | 38.3 | 53.3 | 39.8 | 44.6 | 30.8 | 32.3 |
| **Ours** | **40.3** | **61.0** | **44.0** | **43.8** | **52.5** | **40.0** | **55.1** | **46.5** | **47.1** | **32.1** | **33.2** |

## A.7 QUALITATIVE RESULTS

**Category Frequency.** We present a spider chart in Fig. S1 to illustrate the improvements achieved for various categories. As seen, our method yields substantial performance gains in these categories.

**Class Distribution.** Fig. S3 visualizes the instance distributions with respect to different class categories. While the original dataset (blue) exhibits a severe long-tail bias, our generated data (red) effectively supplements the tail region. This targeted enrichment flattens the overall distribution, confirming that our strategy successfully mitigates class imbalance.

**Visualization.** We provide three sets of visualizations to demonstrate the effectiveness of our approach in layout recalibration, geometry-faithful generation, and debiased object detection. First, Figures S2-S4 illustrate the recalibrated layouts based on representation scores (§3.1), which effectively adjust object positions and sizes as needed, and generate new objects of desired categories. Next, Figures S5-S6 show that our method can generate geometry-faithful images from conditional layouts. In contrast, GeoDiffusion (Chen et al., 2024a) fails to render complex scenes with multiple objects. Finally, these advancements lead to superior detection performance (*i.e.*, Fig. S7), where our method delivers significantly more precise detection results.

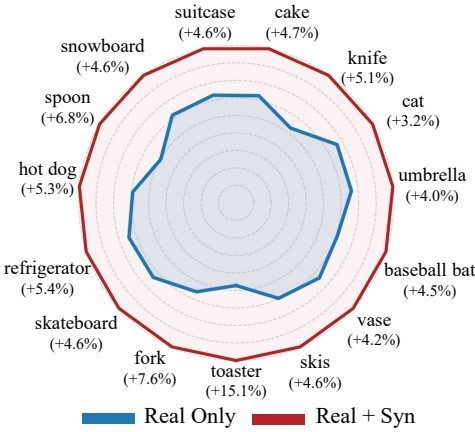

Figure S1: Spider chart illustrating improvements in mAP across various categories.

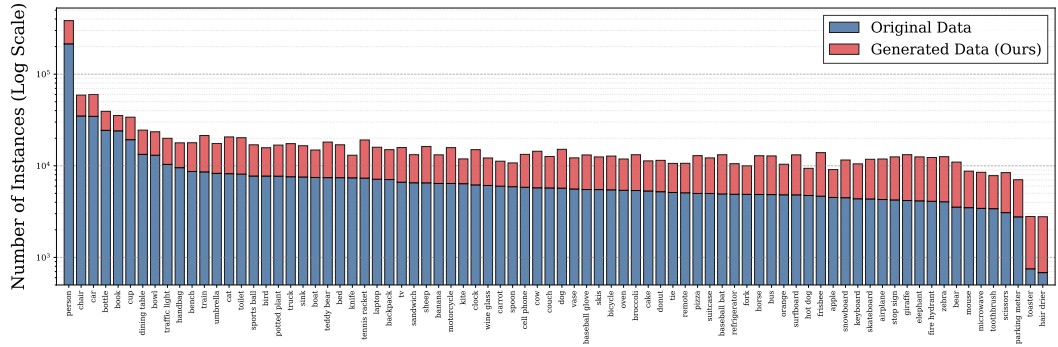

Figure S3: Visualization of class distribution.

---

**Algorithm 1** Visual Blueprint Construction

---

**Require:** Layout set $L = \{(b_i, c_i)\}_{i=1}^{N}$, total classes $N_{cls}$, transparency factor $\alpha$, decrement step $\delta$
**Ensure:** Visual Blueprint $I_{cond}$
 1: $I_{cond} \leftarrow$ zero-initialized image of size $H \times W \times 3$
 2: $counts \leftarrow$ hash map initialized to 0
 3: Sort $L$ based on area of $b_i$ in descending order     *// Handle occlusion: larger objects first*
 4: **for** $i = 1$ to $N$ **do**
 5:     $(b, c) \leftarrow L[i]$
 6:     $h \leftarrow (c + 1)/N_{cls}$     *// Inter-class discrimination via Hue*
 7:     $r \leftarrow counts[c]$
 8:     $v \leftarrow \max(0.2, 1.0 - r \times \delta)$     *// Intra-class distinction via Value decrement*
 9:     $counts[c] \leftarrow counts[c] + 1$
10:     $color \leftarrow$ HSV2RGB$(h, 1.0, v)$
11:     $ROI \leftarrow I_{cond}[b]$
12:     $I_{cond}[b] \leftarrow \alpha \cdot color + (1 - \alpha) \cdot ROI$     *// Occlusion-aware transparency blending*
13: **end for**
14: **return** $I_{cond}$

---

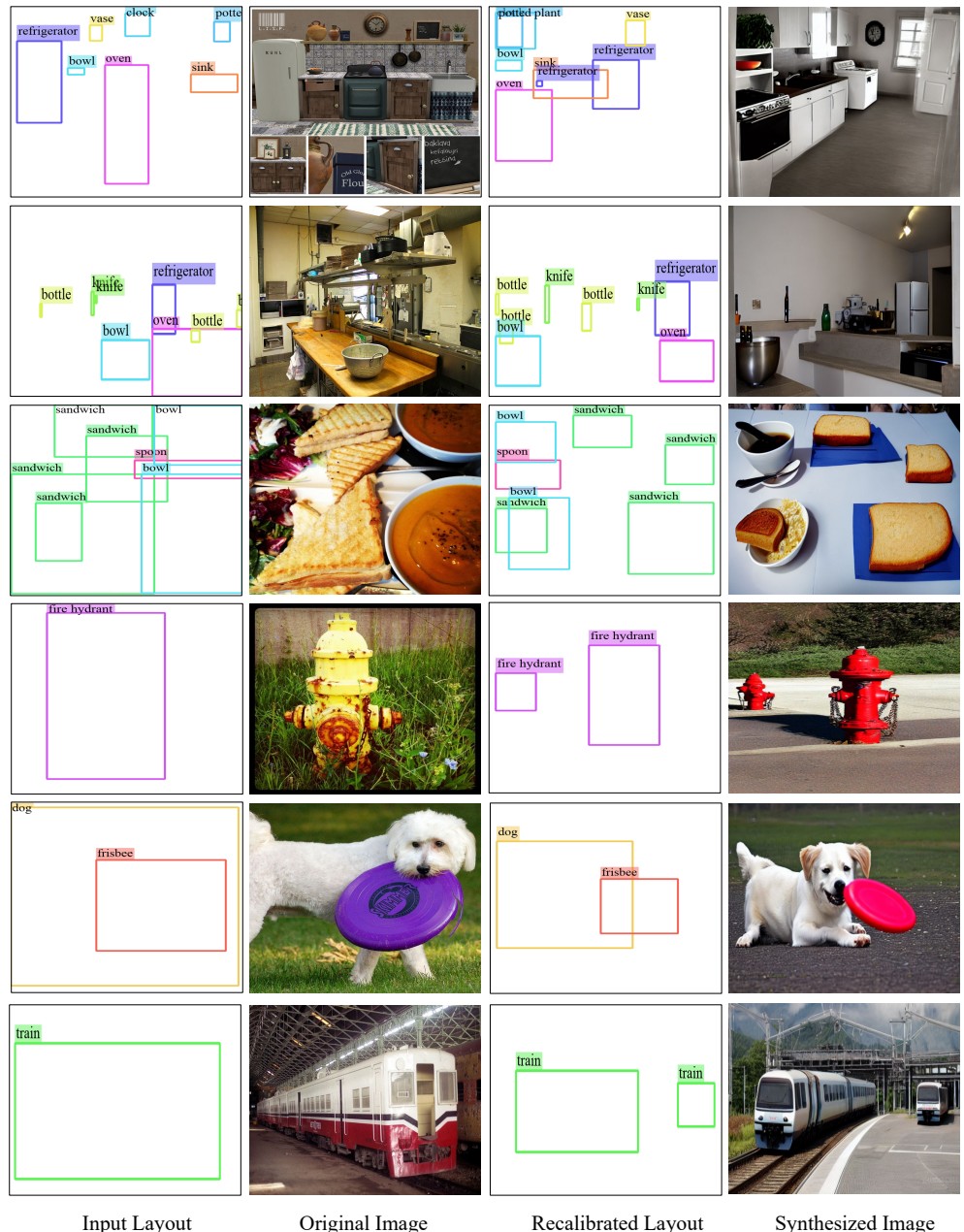

Figure S2: Visualization results for layout recalibration based on representation scores (§3.1) and L2I synthesis using our proposed visual blueprint-prompted method (§3.2).

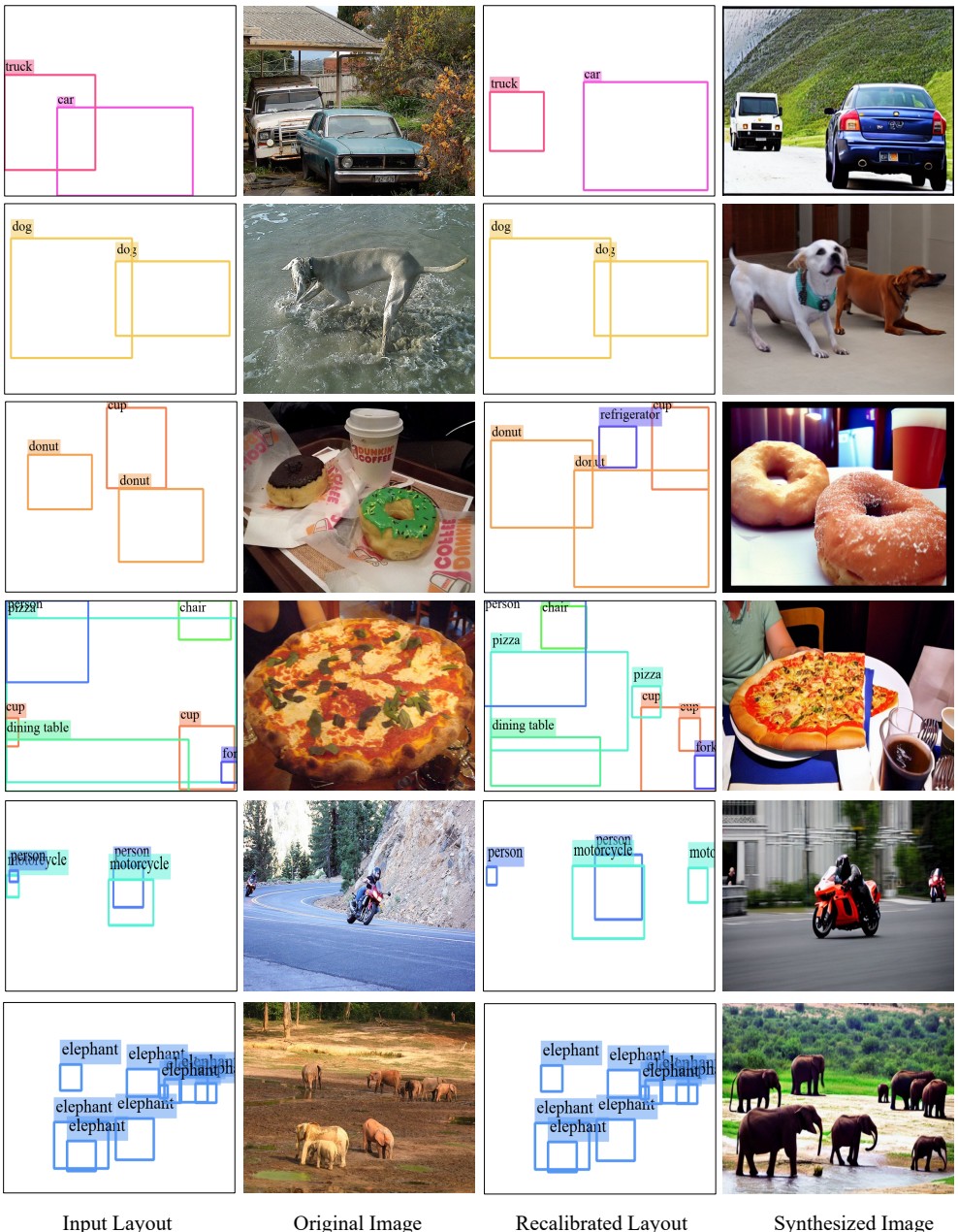

Input Layout     Original Image     Recalibrated Layout     Synthesized Image

Figure S3: Visualization results for layout recalibration based on representation scores (§3.1) and L2I synthesis using our proposed visual blueprint-prompted method (§3.2).

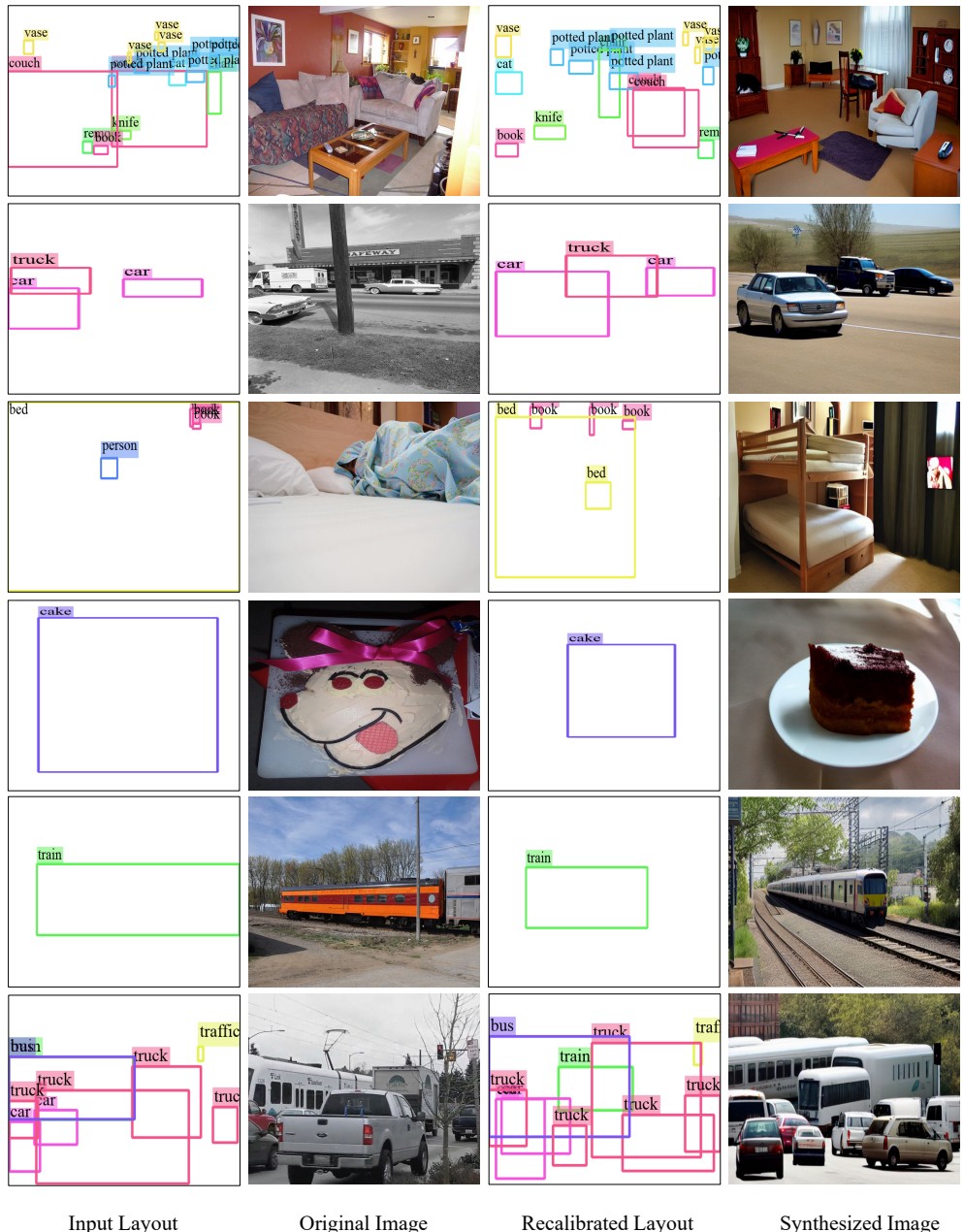

Figure S4: Visualization results for layout recalibration based on representation scores (§3.1) and L2I synthesis using our proposed visual blueprint-prompted method (§3.2).

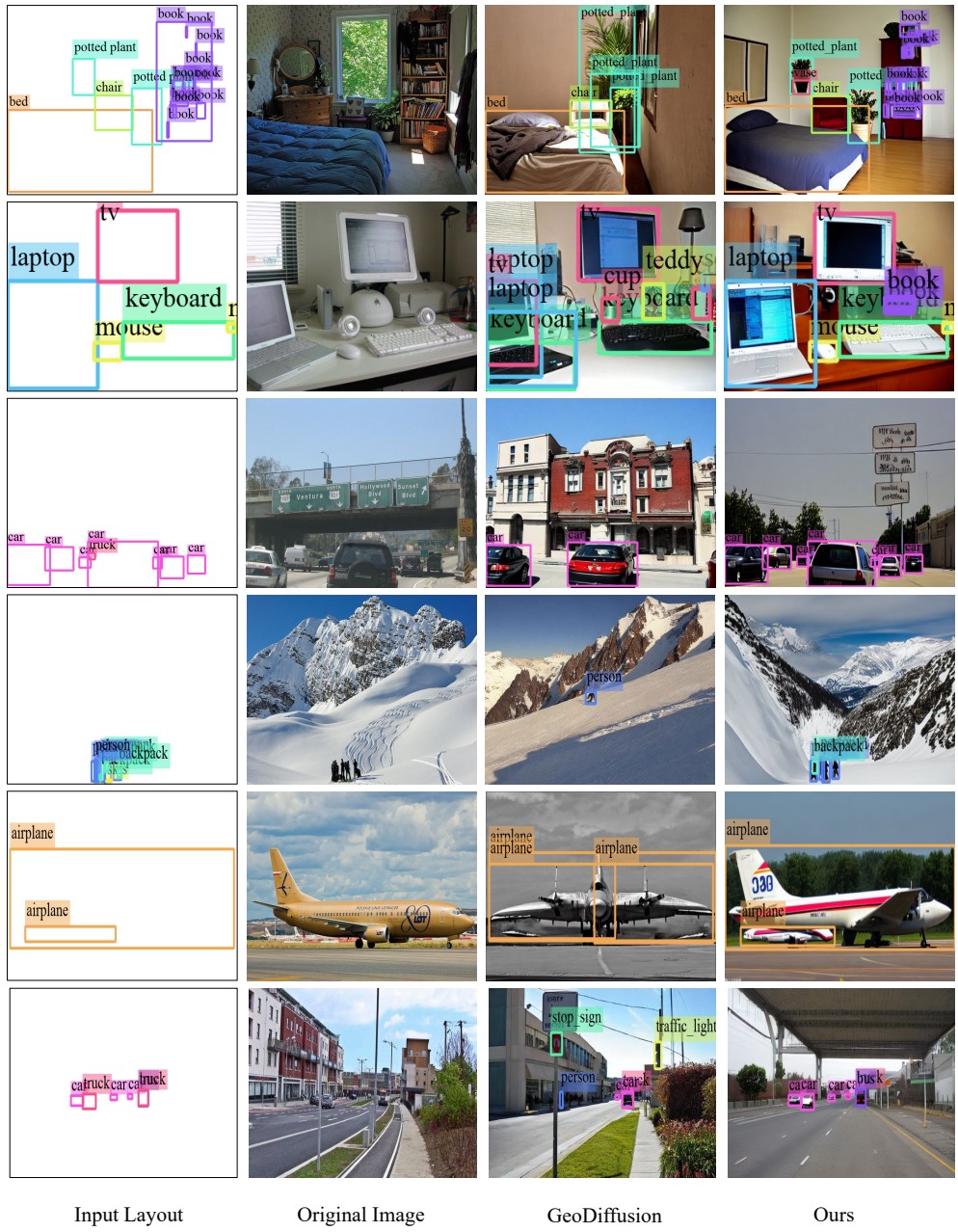

Figure S5: Comparison against GeoDiffusion under the **Fidelity** setup on MS COCO, where the L2I synthesis model should generate geometry-faithful images conditioned on given layouts.

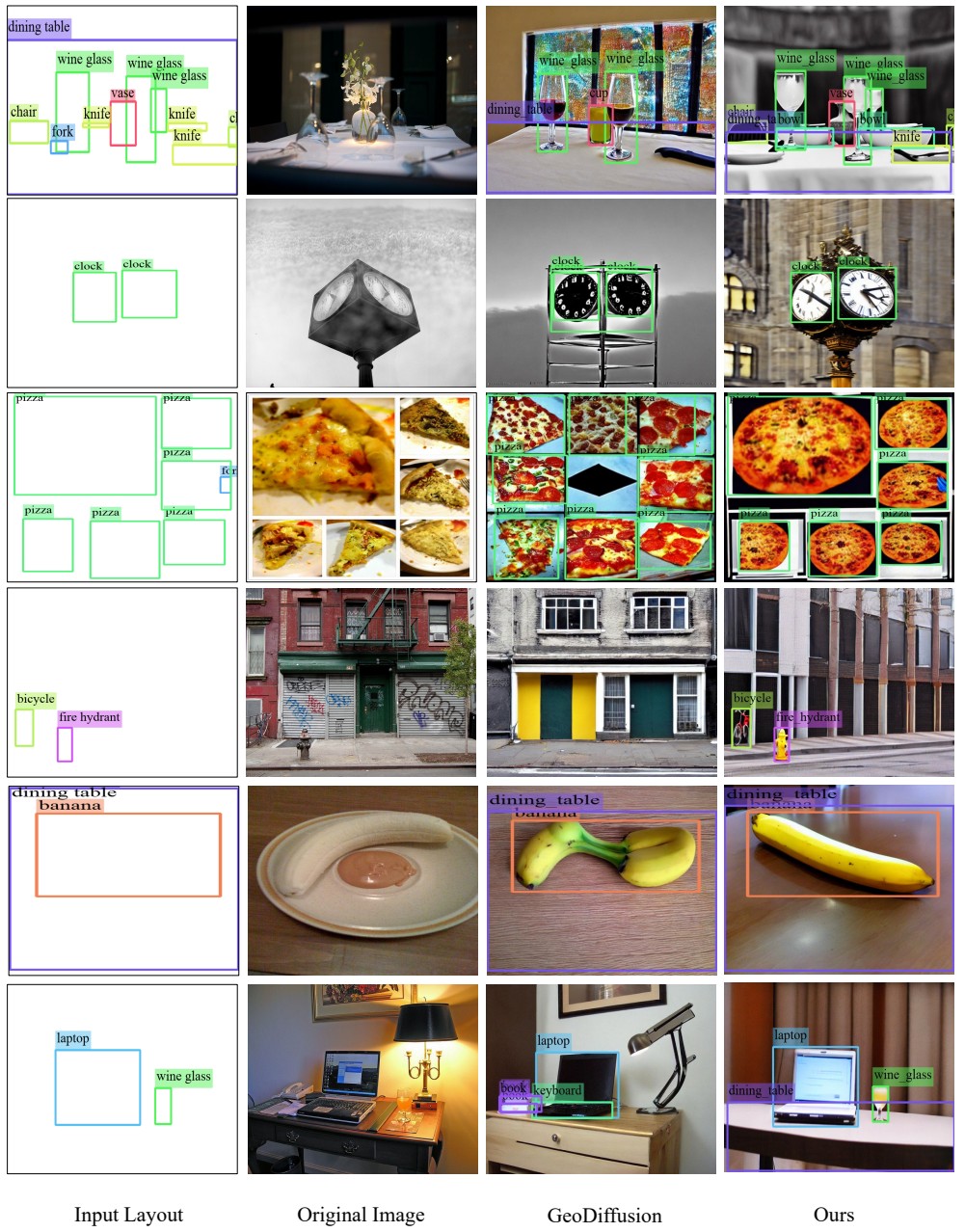

Input Layout        Original Image        GeoDiffusion        Ours

Figure S6: Comparison against GeoDiffusion under the **Fidelity** setup on MS COCO, where the L2I synthesis model should generate geometry-faithful images conditioned on given layouts.

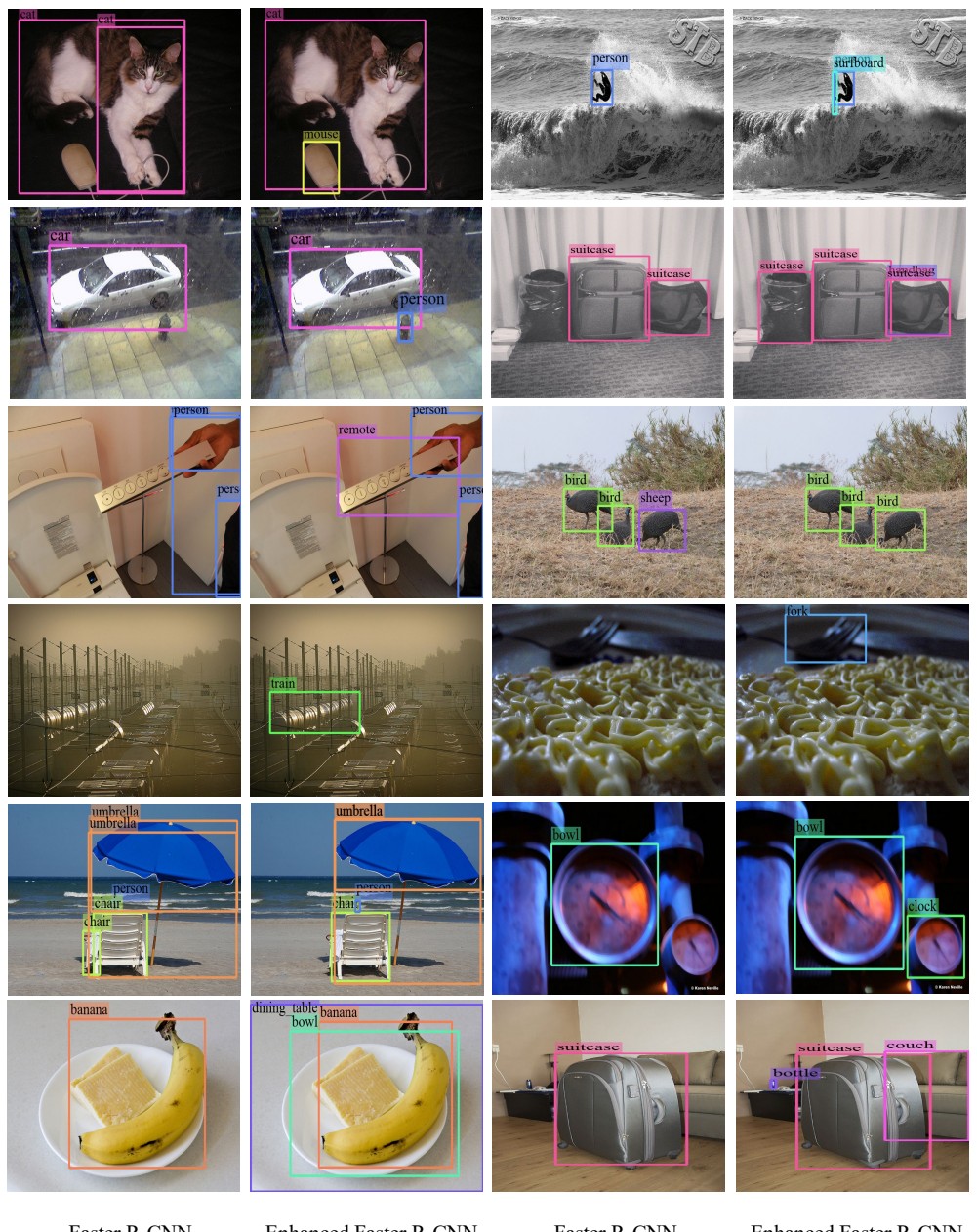

Faster R-CNN     Enhanced Faster R-CNN     Faster R-CNN     Enhanced Faster R-CNN

Figure S7: Visualization results for object detection on MS COCO under the **Debiasing** setup.

