# OpenReview forum: "Unbiased Object Detection Beyond Frequency with Visually Prompted Image Synthesis"
_ICLR.cc/2026/Conference — ICLR 2026 Poster_

### Official Review · Reviewer_d5R9 · 2025-10-27

**Soundness:** 3
**Presentation:** 3
**Contribution:** 3
**Rating:** 4
**Confidence:** 4

**Summary:**

This paper proposes a generation-based debiasing framework for object detection that goes beyond frequency-based methods. The authors introduce a Representation Score (RS) to quantify representation quality by combining instance frequency, visual diversity, and contextual diversity, enabling targeted layout recalibration to synthesize unbiased data. To improve synthesis fidelity, they replace textual layout prompts with visual blueprints that provide unambiguous spatial guidance and propose a duality-aware generative alignment mechanism to align the detector and generator. Experiments on MS COCO and NuImages demonstrate substantial gains in both detection performance and generation fidelity, establishing a new state of the art in debiasing for object detection.

**Strengths:**

1. The motivating study in Section 2 is carefully designed and convincingly demonstrates the “frequency trap” and “fidelity gap.” This provides a solid empirical foundation for the proposed approach.
2. Extensive experiments on MS COCO and NuImages show large and consistent improvements. The ablation studies are thorough, demonstrating the contribution of each component.
3. The proposed framework integrates multiple innovations:
(i) an RS-driven layout recalibration module,
(ii) a visual blueprint-prompted synthesis mechanism, and
(iii) a duality-aware generative alignment strategy.
Together, these components form a well-engineered pipeline with clear rationale and internal consistency.

**Weaknesses:**

1. The experiments focus primarily on Faster R-CNN + ResNet-50. It is unclear whether the proposed method generalizes to other detection paradigms (e.g., DETR, YOLOv11). Low performance caused by data bias may not be significant in these modern models.

2. When the number of categories is large (e.g., 1000 classes), the colors representing different categories in the Visual Blueprint become highly similar, leading to confusion in the generated features. Therefore, compared with layout-based image generation methods, this approach is only suitable for scenarios involving a limited number of categories.

3. While RS integrates frequency, visual, and contextual diversity, the weighting and combination (Eq. 2) seem heuristic. No formal analysis or ablation on hyperparameter sensitivity (e.g., β) is provided.

4. The paper is dense, with long mathematical sections and multiple notations introduced in quick succession. Some figures (e.g., Fig. 2 and Fig. 3) could be clearer in explaining how each component interacts dynamically during training.

**Questions:**

1. Using the Visual Blueprint as the control condition is exactly the same as in ControlNet. So why is the performance of ControlNet + Resampling so poor?

---

> ### Author Response · Authors · 2025-11-30
>
> We thank reviewer d5R9 for the valuable time and constructive feedback. We provide point-to-point response below.
>
> ---
>
> **Q1: Generalization to other detection paradigms.**
>
> **A1:** We thank the reviewer for the suggestion. We have added the quantitative debiasing results for different modern detectors (i.e., YOLOX, DINO, and CO-DETR) in Table 9:
>
> | Detector | mAP (↑)| outer (↑) | rare (↑) | large (↑) | small (↑) |
> |-----|--------|-------|----|-------|--------|
> | YOLOX-s      | 40.5           | 31.2      | 44.5     | 53.1      | 23.5      |
> | + **Ours**   | **43.3** (+**2.8**) | **34.5**      | **48.1**     | **56.2**      | **25.1**      |
> | DINO   | 49.0   | 40.5    | 50.5  | 64.0   | 31.4    |
> | + **Ours**   | **51.8** (+**2.8**) | **44.1**      | **54.2**     | **67.2**      | **33.0**      |
> | CO-DETR      | 52.0    | 43.5      | 53.5     | 67.1   | 34.8      |
> | + **Ours**   | **54.6** (+**2.6**) | **46.8**      | **57.2**     | **70.0**      | **36.3**      |
>
> It can be observed that our method yields consistent and significant gains ($\sim 2.6\text{-}2.8$ mAP), independent of the baseline detector's performance.
>
> **Q2: Scalability to large category sets.**
>
> **A2:** Thanks for your constructive feedback. We would like to clarify that our method remains effective even when the number of categories is large (e.g., 1000 classes). The reasons are given below:
>
> 1. **High-Dimensional Representation Capacity**: Standard layout-based methods typically use simple class maps that represent categories as a 1D integer sequence (e.g., 0, 1, $\dots$, 999). In this low-dimensional space, adjacent classes are numerically proximate and difficult for encoders to differentiate. In contrast, our Visual Blueprint projects these categories into a 3D RGB space ($256 \times 256 \times 256 \approx 16.7$ million possibilities). Even with 1000 classes, distributing them within this high-volume volumetric space yields significantly higher signal variance compared to a scalar map. This allows the encoder to extract distinct features even as the category count increases.
>
> 2. **Intra-class Discrimination & Occlusion-aware Rendering**: Unlike simple binary maps that merge instances, our Visual Blueprint explicitly encodes separation. We assign distinct HSV values to different instances of the same class and render objects in descending size order with transparency. This preserves occlusion patterns and ensures adjacent objects maintain distinct boundaries—even if they belong to the same or color-similar categories—thereby preventing feature confusion regardless of the total category count.
>
> **Q3: Hyperparameter ablation and sensitivity analysis.**
>
> **A3:** The coefficient $\beta$ balances the weight between visual diversity and context diversity. We have added an ablation study in Table S3 of the Appendix:
>
> | $\beta$ Value | 0.0 | 0.5 | 1.0 (Default) | 1.5 | 2.0 |
> |---------|-----|-----|----------------|-----|-----|
> |outer↑   | 30.4 | 30.7 | 30.9 | 30.8 | 30.5 |
> | mAP     | 39.3 | 39.4 | 39.5 | 39.5 | 39.3 |
>
> When $\beta$=0, $\mathcal{D}_\text{ctx}$ is disabled. We use $\beta$=1 as the default setting, and have omitted this hyperparameter in the manuscript.
>
> **Q4: Clarity of figure presentation.**
>
> **A4:** We thank the reviewer for this constructive suggestion. We have added a new Fig. 2 in the revised manuscript to illustrate the complete pipeline, which consists of four integrated stages:
>
> 1. **Bias diagnosis** to analyze real data statistics to compute the Representation Score (RS) (Eq. 2), quantifying dataset biases across frequency and diversity (Eq. 1).
>
> 2. **Layout planner**, which performs RS-Driven Layout Recalibration (Eq. 3-5) to sample target layouts specifically for under-represented groups.
>
> 3. **Blueprint-prompted L2I synthesis** to convert recalibrated layouts into Visual Blueprints (Eq. 9-10), which provide pixel-level conditions for the L2I Generator (Eq. 11-12) to synthesize high-fidelity images.
>
> 4. **Dynamic update mechanism**, which is constrained by Duality-Aware Generative Alignment (Eq. 13) for feature consistency and Error-based Dynamic Debiasing (Eq. 6-7) for adaptive RS updates.
>
> Finally, we use the merged data to re-train a debiased detector (Eq. 14). We hope this figure helps clarify our paper.

---

> ### Author Response · Authors · 2025-11-30
>
> **Q5: Distinction between Visual Blueprint and ControlNet conditioning.**
>
> **A5:** Thank you for your insightful question. While both use spatial conditioning, the fidelity of L2I synthesis differs fundamentally along three axes:
>
> 1. **Inter-class discrimination**: Simple binary maps typically represent classes with adjacent integers, leading to low numerical variance and potential ambiguity for the encoder (e.g., the *Person* class corresponds to (0, 0, 0) and *Sheep* to (0, 0, 19)). These numerically proximate values provide a weak signal for feature extraction. In contrast, we project class labels into the RGB color space by mapping them to equidistant hues on the HSV unit circle (Eq.9-10). This ensures significantly distinct pixel values (e.g., *Person* is rendered as (255, 19, 0) and *Sheep* as (255, 215, 0)), providing a high-variance signal that is much easier for the encoder to discriminate. We have included a **visualization of this comparison** in the Appendix Fig.S1.
>
> 2. **Intra-class (instance) discrimination**: A single image often contains multiple instances of the same category. To enhance intra-class discrimination within an image, we first assign distinct HSV values to different instances of the same class using a small decrement step.
>
> 3. **Occlusion-aware rendering**: We render objects in descending box-size order while applying transparency to background instances, which preserves occlusion patterns and prevents smaller instances from being completely erased.
>
> The differences in L2I synthesis performance can be observed in Tab.6:
>
> | Method              | FID ($\downarrow$) | mAP ($\uparrow$) | AP50 ($\uparrow$) | AP75 ($\uparrow$) |
> |---------------------|:-------:|:-------:|:--------:|:--------:|
> | Baseline            | 28.14   | 25.2    | 46.7     | 22.7     |
> | + Pixel Canvas (Inter-class discrimination)      | 20.15   | 40.8    | 56.2     | 40.5     |
> | + Intra-class (instance) discrimination | 17.05   | 44.5    | 59.5     | 48.8     |
> | + Occlusion-aware rendering    | 15.24   | 46.5    | 61.4     | 51.6     |
>
>
> As seen, all three enhancements deliver considerable improvements on both detection performance and generation fidelity.

---

### Official Review · Reviewer_sxvq · 2025-10-29

**Soundness:** 3
**Presentation:** 3
**Contribution:** 3
**Rating:** 4
**Confidence:** 4

**Summary:**

This paper rethinks the effectiveness of generated data for training object detectors, and proposes the representation score (RS) as a better metric than frequency to select more beneficial layouts, followed by layout recalibration. To enhance the generation quality, the authors propose to utilize the visual blueprint as the task prompt and then design a new loss term to enhance detector-generator dual awareness. Extensive experimental results demonstrate the effectiveness of the proposed method.

**Strengths:**

- The problem setting is fairly clear, including enhancements for both layout and image generation.
- The preliminary exploration of layout frequency is extensive.
- The authors conduct experiments on both fidelity and trainability on both COCO and NuImages.

**Weaknesses:**

- About Representation Score:
  - In line 161, considering that the box size s and horizontal position u are both continuous, do you conduct any quantization to construct the RS group?
  - In Sec. 2, the authors conduct extensive experiments to demonstrate that frequency is not the best metric for layout selection, which, however, cannot directly connect with the complicated definition of RS in Equ. 1.
  - Considering `Freq-Aware Gen` is still a solid baseline, I would expect to see a comparison between it and the complicated RS, with or without the more complicated RS calibration.
- About Visual Blueprints:
  - Besides comparing with the textual prompting of GeoDiffusion, a more direct comparison would be using ControlNet, but the visual prompt is a simple class binary map (i.e., a map of size HxWxC, and the area belonging to class C will be 1 in the C-dimension). It seems that this simple baseline is quite similar to the visual blueprint.
  - Duality-Aware Generative Alignment is proposed to enhance the detector instead of the generator. Do I understand correctly?
- Overall, this paper studies an interesting problem. The first half is good, but the solution is mostly based on intuition. Considering the complexity of the proposed method, a more detailed step-by-step ablation is necessary. I would like to review the author's rebuttal to inform my final decision.

**Questions:**

In Figure 4 (right), 1st row and 2nd column, the `dining table` class name is missing.

---

> ### Author Response · Authors · 2025-11-30
>
> We thank reviewer sxvq for the valuable time and constructive feedback. We provide point-to-point response below.
>
> ---
>
> **Q1: Quantization of box size $s$ and $u$ to construct the RS group.**
>
> **A1:** Thank you for your careful review. We do quantize both box size $s$ and horizontal position $u$ before computing RS. Specifically:
>
> - For box size $s$, we discretize object sizes into small, medium, and large three categories following the standard MS COCO definitions (Small: area $<32^2$; Medium: $32^2 \leq$ area $\leq 96^2$; Large: area $>96^2$).
>
> - For horizontal position $u$, we normalize the horizontal center coordinate to $[0,1]$, and discretize it into $K=10$ uniform bins, to ensure each group $\mathcal{G}$ contains statistically significant samples for RS calculation.
>
> Sec.3 and Appendix A.3 have been updated to avoid confusion.
>
>
> **Q2: Connection between Sec.2 analysis and the RS definition.**
>
> **A2:** We thank the reviewer for the valuable comments. Below, we provide a more detailed explanation of the motivation ($\S$2) and our solution (Eq.1):
>
> - We observed that even high-frequency groups (e.g., large objects) can be under-represented. This indicates relying solely on frequency ($\mathcal{D}_{freq}$) fails to explain this.
>
> - Therefore, we design RS which aims to measure “how well a group is represented” rather than “how often it appears”. We introduce Eq.1 to quantify this "representation density" from two complementary perspectives:
>
>     1. $\mathcal{D}_{vis}$ (Intra-group Visual Variation): measures the spread of feature embeddings. It ensures that we generate more data for groups that are visually diverse (high variance), even if they are frequent.
>
>     2. $\mathcal{D}_{ctx}$ (Context Co-occurrence): measures the richness of semantic surroundings. It ensures we target groups that appear in limited contexts (low co-occurrence), preventing the model from relying on background shortcuts.
>
> We have updated the detailed explanation of the evidence and the solution in $\S$3.
>
>
> **Q3: Quantitative Comparison between Freq-Aware Gen and RS.**
>
> **A3:** In the earlier version of the paper, we already included this specific ablation in Table 8, and we apologize for the unclear labeling. In the updated version, we have added more detailed ablation studies:
> | Score                                      | mAP↑ | outer↑ | rare↑ | large↑ | small↑ |
> |----------|:----:|:------:|:-----:|:------:|:------:|
> | Bias-Agnostic                             | 39.1 |  29.9  | 45.2  |  51.3  |  22.1  |
> | $\mathcal{D}_\text{freq}$                 | 39.3 |  30.4  | 45.8  |  50.9  |  22.5  |
> | $\mathcal{D}_\text{freq+vis}$ | 39.7 |  30.7  | 46.3  |  52.0  |  22.6  |
> | $\mathcal{D}_\text{freq+ctx}$ | 39.5 |  30.9  | 46.1  |  51.2  |  22.7  |
> | $\mathcal{D}_\text{vis+ctx}$  | 39.5 |  30.6  | 45.9  |  51.7  |  22.4  |
> | $\mathcal{D}_\text{freq+vis+ctx}$ | 39.9 |  31.0  | 46.4  |  52.3  |  22.8  |
>
> Note that **freq** denotes the ''Freq-Aware Gen'' baseline,
> and **freq**+**vis**+**ctx** denotes the full RS method.
>  It can be observed that relying solely on frequency **freq** yields limited gains, particularly for large objects. Incorporating **vis** effectively addresses this by capturing intra-class variance, significantly boosting large object performance (+1.1 AP). Meanwhile, **ctx** helps diversify contexts, improving performance in outer regions (+0.5 AP). Consequently, our final method integrates all components to achieve the best overall performance.

---

> ### Author Response · Authors · 2025-11-30
>
> **Q4: Visual Blueprints vs. ControlNet (Simple Class Binary Map).**
>
> **A4:** We agree that ControlNet is an important baseline. In the original manuscript, we compare our method against ControlNet in Table 1 (Generation Fidelity) and Table 2 (Detection Debiasing), where our method significantly outperforms ControlNet by **11.1** AP in generation and **3.6** mAP in detection.
> These gains arise primarily because our proposed Visual Blueprint differs from ControlNet along three key axes:
>
> 1. Inter-class Discrimination. The binary maps in ControlNet typically represent classes with adjacent integers. This leads to low numerical variance and potential ambiguity for the encoder (e.g., the *Person* class corresponds to (0, 0, 0) and *Sheep* to (0, 0, 19)). These numerically proximate values provide a weak signal for feature extraction. In contrast, we project class labels into the RGB color space by mapping them to equidistant hues on the HSV unit circle (Eq. 9-10). This ensures significantly distinct pixel values (e.g., *Person* is rendered as (255, 19, 0) and *Sheep* as (255, 215, 0)), providing a high-variance signal that is much easier for the encoder to discriminate. We have included a **visualization of this comparison** in the Appendix Fig. S1.
>
> 2. Intra-class Discrimination. A single image often contains multiple instances of the same category. To enhance intra-class discrimination within an image, we first assign distinct HSV values to different instances of the same class using a small decrement step.
>
> 3. Occlusion-aware Rendering. We render objects in descending box-size order while applying transparency to background instances, which preserves occlusion patterns and prevents smaller instances from being completely erased.
>
> Additionally, we have updated ablative studies of Visual Blueprint design components in Tab. 6, where our design consistently delivers considerable improvements.
>
> | Method              | FID ($\downarrow$) | mAP ($\uparrow$) | AP50 ($\uparrow$) | AP75 ($\uparrow$) |
> |---------------------|:-------:|:-------:|:--------:|:--------:|
> | ControlNet (binary maps) | 28.14   | 25.2    | 46.7     | 22.7     |
> | + Pixel Canvas      | 20.15   | 40.8    | 56.2     | 40.5     |
> | + Instance Discrim. | 17.05   | 44.5    | 59.5     | 48.8     |
> | + Overlap Aware.    | 15.24   | 46.5    | 61.4     | 51.6     |
>
>
>
>
> **Q5: Duality-Aware Generative Alignment enhances the Detector.**
>
> **A5:** The mechanism indeed serves to enhance the Detector.  By adding the alignment loss $\mathcal{L}_{image}^{IA}$, we penalize the detector if it extracts features from a synthesized image that can not reconstruct the original layout. This forces the detector to be robust to the features produced by the generator.
> We have provided further clarification in the revised version.
>
>
>
> **Q6: Figure 4 missing label.**
>
> **A6:** Thank you for spotting this. We have updated Fig. 4.

---

### Official Review · Reviewer_yyBg · 2025-11-02

**Soundness:** 3
**Presentation:** 2
**Contribution:** 3
**Rating:** 6
**Confidence:** 4

**Summary:**

This paper proposes a generation-based debiasing framework for object detection. It claims that frequency is an incomplete proxy and previous methods lacks fidelity. Then it designs a representation score to replace frequency, this score can quantify how well a concept is represented across both sample density and representation diversity. This paper also presents a visual blueprint to replace text prompts so that offer a clear instruction to improve the fidelity.

**Strengths:**

1. The performance is good. The improvement is substantial compared to previous methods, such as GeoDiffusion.
2 . This method is simple yet effective. The two main contributions, the representation score and visual blueprint, are easy to understand and significantly improve the performance.
3. The motivation is clear. The observations in Sec. 2 explain why frequency is not enough and fidelity is important.

**Weaknesses:**

1. Lack a figure to describe the overall pipeline in detail. Figures 2 and 3 are used to explain the layout recalibration and blueprint construction; however, there is no figure to explain the complete pipeline, which can be confusing.
2. The effect of Generative Alignment is negligible. Can you explain its necessity?
3. There is no connection between the two contributions in this paper, which makes the paper an incremental industry-focused work rather than a cohesive academic study.

**Questions:**

See weakness above.

---

> ### Author Response · Authors · 2025-11-30
>
> We thank reviewer yyBg for the valuable time and constructive feedback. We provide point-to-point response below.
>
> ---
>
> **Q1: Lack of a figure to describe the overall pipeline in detail.**
>
> **A1:** We thank the reviewer for this constructive suggestion. We have added a new Fig. 2 in the revised manuscript to illustrate the complete pipeline, where our framework consists of four stages:
>
> 1. **Bias diagnosis** to analyze real data statistics to compute the Representation Score (RS) (Eq. 2), quantifying dataset biases across frequency and diversity (Eq. 1).
>
> 2. **Layout planner**, which performs RS-Driven Layout Recalibration (Eq. 3-5) to sample target layouts specifically for under-represented groups.
>
> 3. **Blueprint-prompted L2I synthesis** to convert recalibrated layouts into Visual Blueprints (Eq. 9-10), which provide pixel-level conditions for the L2I Generator (Eq. 11-12) to synthesize high-fidelity images.
>
> 4. **Dynamic update mechanism**, which is constrained by Duality-Aware Generative Alignment (Eq. 13) for feature consistency and Error-based Dynamic Debiasing (Eq. 6-7) for adaptive RS updates.
>
> Finally, we use the merged data to re-train a debiased detector (Eq. 14). We hope this figure helps clarify our paper.
>
> **Q2: The effect of Generative Alignment.**
>
> **A2:** Sorry for this confusion. Duality-Aware Generative Alignment serves to i) prevent overfitting or shortcuts, and ii) improve training stability.
>
> In generative data augmentation, synthetic data tends to introduce stochastic artifacts (e.g., unstable textures) that vary with random seeds. Without constraint, the detector may overfit to these artifacts, and lead to high variance across different training runs. The Alignment Loss ($\mathcal{L}_\text{image}^\text{IA}$) serves as a structural anchor, which enforces feature consistency and reduces the sensitivity to generative noise. We conduct two experiments to verify this.
>
> **First**, we measured the feature consistency between real and synthesized images. We extracted RoI features from the detector trained with and without $\mathcal{L}_\text{image}^{IA}$ on a paired set of real and synthesized samples, which shares identical layouts.
>
> As shown, adding $\mathcal{L}_{image}^{IA}$ boosts the similarity from 0.47 to 0.76. This proves that the alignment loss forces the detector to learn physically consistent representations, and filter out high-frequency generative noise.
>
> | Setup | w/o $\mathcal{L}_\text{image}^{IA}$ | w $\mathcal{L}_\text{image}^{IA}$ |
> |-----|-----|-----|
> | Similarity | 0.47 | 0.76 |
>
>
>
> Second, we give three independent training runs of the detector with different random seeds for both settings, and report the mean mAP and the standard deviation (std) to quantify stability. As seen below, with generative alignment, the std decreases from 0.32 to 0.06, verifying its ability to stabilize the training process.
>
>
> | Method                | Mean mAP ($\uparrow$) | std ($\downarrow$)|
> |-----------------------|----------|--------------------------|
> | w/o $\mathcal{L}_\text{image}^{IA}$         | 40.1     | 0.32      |
> | w $\mathcal{L}_\text{image}^{IA}$| 40.3| 0.06       |
>
>
> Both experiments have been added to the Appendix.

---

> ### Author Response · Authors · 2025-11-30
>
> **Q3: The connection between two contributions (Scoring-Driven Debiasing & Visual Blueprint).**
>
> A3: We thank the reviewer for the valuable feedback. The two contributions in this paper are interdependent, where the Scoring-Driven Debiasing identifies **what to generate**, and the Visual Blueprint provides the necessary fidelity to determine **how to generate** it faithfully:
>
> 1. **What to generate**: Unlike standard frequency-based mining, our Scoring-Driven Debiasing Engine utilizes a multi-dimensional Representation Score to quantify latent diversity gaps. It actively constructs **debiased layout distributions** targeting specific missing contexts, which standard generative pipelines struggle to address.
>
> 2. **How to generate**: As shown in our Motivating Study ($\S$2), standard (i.e., text-prompted) L2I generators suffer from a severe fidelity gap. If we rely solely on **what to generate**, the recalibrated layouts would be rendered poorly due to spatial ambiguity. This would produce low-fidelity images that fail to debias and improve the detector.
>
> This dependency is empirically supported by comparing the impact of our debiasing strategy on a text-prompted (i.e., GeoDiffusion) versus our proposed Visual Blueprint generator:
>
> | Method                                   | mAP    |
> |------------------------------------------|--------|
> | baseline                                 | 37.4   |
> | GeoDiffusion                             | 38.4   |
> | GeoDiffusion + Scoring-Driven Debiasing  | 38.7 (+**0.3**) |
> | Visual Blueprint                         | 38.9   |
> | Visual Blueprint + Scoring-Driven Debiasing | 40.3 (+**1.4**) |
>
> As seen, when applying our debiasing strategy to a standard generator (GeoDiffusion), the gain is minor (+0.3 mAP) as the generator cannot generate layout-faithful images for debiasing. In contrast, pairing it with our Visual Blueprint unlocks a significant **+1.4 mAP** improvement. This confirms the interdependence between the two contributions.
>
> We have added the experiment and the analysis above in Table S2 of the Appendix.

---

### Official Review · Reviewer_8Zww · 2025-11-02

**Soundness:** 3
**Presentation:** 3
**Contribution:** 3
**Rating:** 6
**Confidence:** 5

**Summary:**

This paper presents a novel debiasing framework for object detection, named "Beyond Frequency: Scoring-Driven Debiasing for Object Detection via Blueprint-Prompted Image Synthesis." The key contributions are:
Representation Score (RS): A new metric that goes beyond mere frequency counts to diagnose representational gaps in the training data by considering both sample density and representation diversity.
Visual Blueprint-Prompted Image Synthesis: A method that replaces ambiguous text prompts with precise visual blueprints to guide the generation of high-quality, unbiased images.
Generative Alignment Strategy: A mechanism that fosters communication between the detector and generator, ensuring high-fidelity synthesis of debiased samples.
Dynamic Debiasing Engine: A system that continuously refines the RS based on detector errors, ensuring adaptive and targeted debiasing throughout training.
The method significantly improves the performance for underrepresented object groups and achieves state-of-the-art results in terms of both debiasing effectiveness and generation fidelity.

**Strengths:**

1. The paper introduces a novel approach to diagnosing and addressing dataset biases by integrating representation scores with visual blueprints and generative alignment. This method is innovative as it overcomes limitations found in existing techniques.
2. The research is characterized by rigorous experiments and comprehensive analysis. The proposed methods are well-implemented, validated, and demonstrate excellent performance improvements.
3. The contributions are significant as they tackle a major challenge in object detection by providing a practical solution that enhances model performance and fairness.

**Weaknesses:**

I appreciate the great efforts for this paper with clear motivation, thoughtful analyses, detailed strategies and comprehensive evaluations. Even so, after carefully considering the contributions of this work, I have some main concerns on the insights the paper conveys, besides some concerns on details in the paper.
1. *The insights*
  - a) The study for the motivation in Section is too empirical and heuristic. Those observation and analyses mainly focus on the performance comparison. The performance gains/gap motivates this work. This idea is somewhat reasonable. But, it is strongly suggested to provide deeper analyses with qualitative evaluation. E.g., do object detectors need those generated images that are not always well generated by models? What fresh impacts or findings those generated images would bring to detectors, in comparison with conventional augmentation methods? What generated images are more valuable to object detectors?

  - b) Because of a), although this paper is well written, it would be an unconscious misguidance to the main idea proposed in this paper: the layout generation and L2I generation. The core work of this paper is to focus on the L2I generation and there is very few discussions on the object detectors, e.g., only “duality-aware generative alignment” returns to the main task-object detection. If so, why not clearly introduce the work of L2I? Overall, I) I appreciate the whole work, but I sincerely suggest that we can make a different effort for more insights and more essential findings on the research topic, not just performance gap or gains that motivate us. II) For the concerns aforementioned, it is not easy to response or well address in this paper. But, if the authors have more insights or analyses (e.g., mentioned in a)), we can discuss in the rebuttal stage.

 *Concerns on technical details:*

2. The paper could benefit from a more detailed analysis of the sensitivity of the representation score's parameters (e.g., βin RS(G) = Dfreq(G) · (Dvis(G) + β · Dctx(G)) .). Understanding how these parameters affect the results would provide more insight into the robustness of the method.

3. The paper does not discuss the computational cost of the proposed methods. Given the complexity of the generative models and the dynamic debiasing engine, it would be useful to include an analysis of the computational requirements and potential trade-offs.

3.	For the layout generation, it is hard to ensure the accurate and plausible generation results. So layout recalibration is important. But, the strategy in Line180-185 and even the part of “Layout recalibration” worth more considerations.

4.	The layout generation and L2I generation is not always perfect. It is evitable that some negative impacts they make. Thus, it is strongly suggested that those impacts and analyses can be truly provided and analyzed. Those analyses would be a valuable suggestion or lesson for other researchers.

5.	Some details are suggested to provide. E.g., 1) Line263-267: it is kind to explain the more details of the implementation for the reproduction.

**Questions:**

1. What is the computational cost of the proposed methods compared to existing techniques? Are there any trade-offs in terms of training time or resource requirements?
2. Could the authors provide visualizations of the class distributions in both the original and generated datasets? This would help in understanding how the proposed method alters the distribution to achieve debiasing.
3. Could the authors provide more details on how the parameters of the representation score (e.g., β, τ) were chosen? Were any experiments conducted to analyze their sensitivity?

---

> ### Author Response · Authors · 2025-11-30
>
> We thank reviewer 8Zww for the valuable time and constructive feedback. We provide point-to-point response below.
>
> ---
>
> **Q1: The study for the motivation in Section 2 is empirical and heuristic.**
>
> **A1:** We thank the reviewer for this insightful comment. The study in $\S$2 serves as a critical role to reveal the counter-intuitive limitations of current assumptions, e.g., "frequency is the exact proxy for what data need" (Frequency Trap) and "text-to-image synthesis provides good samples" (Fidelity Gap). These empirical findings were necessary to identify where current methods fail.
>
> However, we agree that performance gains alone are insufficient to explain why the proposed method works. In response to the reviewer's suggestion for deeper qualitative analysis, we have summarized three core insights regarding the value of generated data, addressing why imperfect generated images benefit detectors (Q1.1) and what unique value they bring compared to traditional augmentation (Q1.2).
>
> **Q1.1: Do object detectors need those generated images that are not always well generated by models?**
>
> **A1.1:** Thank you for your insightful question. Generated images offer unique value beyond traditional augmentation over **bias mitigation**, **visual consistency**, and **high-dimensional continuit** three aspects:
>
> 1. Real datasets often suffer from contextual correlations (e.g., boats are almost universally found on water). This causes models to overfit to background shortcuts. While generated images may have minor noise, they decouple objects from fixed backgrounds by placing them in diverse contexts. This forces the detector to learn robust object features rather than overly emphasizing background correlations.
>
> 2. Conventional methods like Copy-Paste often introduce sharp edges and inconsistent lighting. This creates artificial artifacts that detectors can overfit to. In contrast, our blueprint-prompted synthesis maintains physical coherence. The generative model can maintain accurate lighting interactions, shadows, and boundary blending. This provides the detector with valuable signals on how objects naturally interact with their environment, which is missing in Copy-Paste augmentation.
>
> 3. Real data represents sparse points in the high-dimensional feature space [1]. While traditional methods can create new samples, they are "constrained by the visual vocabulary of the original dataset" (L34-35). In contrast, generated images are sampled from the observed data distributions, and serve to interpolate the manifold between these points. They act to fill representational gaps, preventing the model from memorizing limited training examples and improving generalization.
>
> [1]. High-dimensional data analysis: The curses and blessings of dimensionality. AMS math challenges lecture, 2000.
>
>
>
> **Q1.2: What fresh impacts or findings those generated images would bring to detectors, in comparison with conventional augmentation methods?**
>
> **A1.2:** Thank you for your question. While conventional methods are limited to "recombining" existing pixels, our generative approach focuses on "re-rendering" and "expanding" the data distribution:
>
> 1. Conventional methods like Copy-Paste are context-agnostic and introduce composite artifacts. They often place objects in semantically impossible locations (e.g., pasting a car onto a sea surface) and create sharp edges. This introduces pixel statistic discontinuities that do not exist in real images. Consequently, detectors may overfit to these artificial boundary cues rather than learning robust object features.
> **In contrast**, our generative approach ensures **semantic and visual coherence**. First, it enforces semantic coherence  (e.g., keeping cars on roads). Second, it performs seamless re-rendering by synthesizing natural object-background interactions (e.g., natural shadows and edges). The detector thus could focus on the object structure rather than artifacts.
>
> 2. Our empirical study in Sec.2 reveals an insight that "Instance frequency is an incomplete proxy for data need" (L120-129). Conventional debiasing methods (e.g., Resampling, Re-weighting) are frequency-centric. They rebalance the dataset by simply duplicating or up-weighting rare samples.
> **In contrast**, our method transforms debiasing from **frequency** to **representation**. Instead of blindly increasing the count of rare classes, we use the Representation Score (RS) to identify specific "Representational Gaps" — areas in the feature space where data is visually or contextually sparse, beyond its frequency count, and then actively synthesize samples to fill these specific voids. This debiases the model against latent diversity shifts, where traditional augmentation methods achieve limited effects.

---

> ### Author Response · Authors · 2025-11-30
>
> **Q1.3: What generated images are more valuable to object detectors?**
>
> **A1.3:** Based on our diagnostic experiments ($\S$2) and the proposed framework, we define "data value" not by quantity (how many images are added), but representational diversity and geometric fidelity. Specifically, the valuable images share two characteristics:
>
> 1. Our observation 2 ("The Frequency Trap") reveals that value is not solely determined by class rarity. Even high-frequency groups (e.g., large objects) can be "data hungry" if their representation is monotonous. Therefore, valuable images are those that populate the sparse regions of the feature manifold, i.e., samples that increase **Visual Diversity** ($D_\text{vis}$) and **Contextual Diversity** ($D_\text{ctx}$). These images provide insights that help the model generalize beyond the limited visual vocabulary of the training set.
>
> 2. The value of generated data is also defined by the "Fidelity". An image is more valuable if its pixel content is **i)** visually coherent with no discontinuity like sharp edges, and **ii)** faithfully respects the semantic layout (i.e., the bounding box actually contains the object). For **i)**, the powerful generation ability of diffusion models can address this challenge.
> For **ii)**, as shown in our comparison between Bias-Agnostic Gen and Real Data, generated images with spatial ambiguity may act as noisy labels that harm training. Thus, valuable images should be geometry faithful with precise control (via **Visual Blueprints**) to ensure that the synthetic data serves as a valid training signal.
>
>
>
>
> **Q2: Detailed analysis of the sensitivity of the representation score's parameters.**
>
> **A2:** Apologies for this missing ablation study. The coefficient $\beta$ in Eq.2 balances the weight between visual diversity and context diversity. We have added an ablation study in the Appendix Tab.S3:
>
> | $\beta$ Value | 0.0 | 0.5 | 1.0 (Default) | 1.5 | 2.0 |
> |---------|-----|-----|----------------|-----|-----|
> |outer↑   | 30.4 | 30.7 | 30.9 | 30.8 | 30.5 |
> | mAP     | 39.3 | 39.4 | 39.5 | 39.5 | 39.3 |
>
> When $\beta$=0, $\mathcal{D}_\text{ctx}$ is disabled. We use $\beta$=1 as the default setting, and have omitted this hyperparameter in the manuscript.
>
> **Q3: Computational cost of the proposed methods.**
>
> **A3:** We thank the reviewer for pointing this out. The total computational cost consists of three parts:
>
> 1. Debiasing Engine: The Bias Diagnosis Engine (computing Representation Scores) and Layout Planner (Recalibration) are computationally lightweight with **merely numerical calculation**, adding negligible overhead to the standard training pipeline.
>
> 2. Blueprint-Prompted L2I Synthesis: This module comprises the Layout Renderer and the L2I Generator.
>
> - The Layout Renderer is extremely lightweight, involving simple **pixel-wise mapping** operations to convert layout boxes into the Visual Blueprint with virtually zero cost.
>
> - For the L2I Generator, we employ an adapter-based fine-tuning strategy (injecting features into a frozen U-Net), rather than full model fine-tuning.
> As a result, training the L2I Generator on MS COCO at a resolution of $512 \times 512$ takes 74 GPU hours, which consumes significantly fewer training times than prior generation-based SOTA.
>
> | Method        | Epoch | GPU Hours |
> |---------------|:-----:|:---------:|
> | Geodiffusion  |  60   |   640     |
> | Ours    |  14   |    74     |
>
> **Q4: The layout recalibration strategy worth more considerations.**
>
> **A4:** We agree with the point that generating layouts from scratch is unstable and often leads to physically implausible scenes (e.g., flying cars). This is precisely why we adopted a recalibration strategy for Layout Recalibration, rather than purely random generation. Our strategy ensures plausibility through four constraints:
>
> 1. **Real-World Anchoring**: We utilize a real layout from the training set as an anchor. This ensures that the generated scene inherits valid physics (e.g., the perspective, vanishing points, and ground plane logic) directly from real-world data.
>
> 2. **Context-Aware Class Selection**: As defined in Eq. 5, our sampling policy for the target class $c'$ is context-aware. It conditions the selection on the set of existing classes $\mathcal{K}$ in the anchor image. This prioritizes objects that are semantically compatible with the current scene, thereby preventing implausible combinations.
>
> 3. **Coupled Attribute Sampling**: We treat size ($s$) and horizontal position ($u$) as coupled attributes rather than independent variables. This preserves the correlation between object scale and spatial depth, maintaining perspective consistency.
>
> 4. **Vertical Constraint**: As detailed in Eq. 4, we apply only a "small Gaussian jitter" to the vertical position ($v$). This constraint preserves the semantic layering of the scene, ensuring the recalibrated layout remains physically plausible.

---

> ### Author Response · Authors · 2025-11-30
>
> **Q5: Analysis of negative impacts from imperfect generation.**
>
> **A5:** We appreciate this constructive suggestion. In our study, we identified two primary types of negative impacts and designed specific modules to mitigate them:
>
> 1. As noted in our "Fidelity Gap" analysis (Observation 3 in $\S$2), standard text-prompted L2I models often **fail to render the exact shape specified by the layout**.  This acts as noisy labels (i.e., the ground-truth box exists, but the visual object is missing or distorted), which confuses the detector and degrades localization performance.
>
>  - To mitigate this, we propose **Visual Blueprint**. By projecting class labels into high-contrast RGB space (HSV mapping), we provide the generator with unambiguous pixel-level guidance.
>
> - As shown in Table 6, replacing textual layouts with our Visual Blueprint significantly improves generation quality, reducing the FID score from 28.14 to 15.24. Moreover, in Table 1, our method achieves a detection mAP of 36.8 on synthesized images, surpassing the prior SOTA (GeoDiffusion) by +15.9 mAP. This confirms that our method generates objects that are more recognizable and geometrically aligned with the provided boxes.
>
> 2. Synthetic data inevitably contains artifacts (e.g., specific noise patterns or high-frequency texture anomalies) inherent to the diffusion process. If left unconstrained, the detector risks **overfitting to these generative artifacts**.
>
> - We address this via **Duality-Aware Generative Alignment** ($\mathcal{L}_{image}^{IA}$). This penalizes the detector if it extracts features from a synthesized image that cannot be used to reconstruct the original layout. It acts as a consistency filter, which forces the detector to ignore generative artifacts and focus on consistent features.
>
> - To verify this, we measured the feature consistency between real and synthesized images, by extracting RoI features from the detector trained with and without $\mathcal{L}_{image}^{IA}$ on a paired set of real and synthesized samples, which shares identical layouts.
>
> - As shown below, the model trained *without* alignment shows a low feature cosine similarity ($0.47$) between real and synthetic pairs.
> In contrast, adding $\mathcal{L}_{image}^{IA}$ boosts the similarity to $0.76$. This quantitatively proves that the alignment loss forces the detector to learn physically consistent representations, and filter out high-frequency generative noise.
>
> | Setup | $\textit{w/o}$ $\mathcal{L}_{image}^{IA}$ | $\textit{w}$ $\mathcal{L}_{image}^{IA}$ |
> |-----|-----|-----|
> | Similarity | 0.47 | 0.76 |
>
>
>
> **Q6: Implementation details for reproduction.**
>
> **A6:** We thank the reviewer for this reminder. To facilitate exact reproduction, we have added **Algorithm 1 (Visual Blueprint Construction)** to Appendix in the revised manuscript, which formally defines the rendering logic.
>
> For quick reference, the specific hyperparameters and logic used in our implementation are:
>
> 1. Color Space: We utilize the HSV color space for maximum separability.
>
> 2. Hue Mapping (h): Classes are mapped to equidistant hues $H \in [0, 1]$ using $h = (c + 1) / N_{cls}$.
>
> 3. Instance Distinction ($\delta$): To distinguish overlapping instances of the same class, we decrement the Value ($V$) by a step $\delta = 0.02$ for each subsequent instance ($v = \max(0.2, 1.0 - r \times \delta)$).
>
> 4. Transparency ($\alpha$): We apply an alpha blending factor of $\alpha = 0.9$ to background objects.
>
> 5. Occlusion Handling: As detailed in Algorithm 1, objects are sorted and rendered in descending order of area size to ensure smaller objects are not occluded by larger ones.
>
> **Q7: Visualizations of the class distributions.**
>
> **A7:** Thanks for your constructive suggestion. We have added a visualization of the instance distributions in Appendix Figure S3.

---

### Author Response · Authors · 2025-11-30
**Summary of response for quick review**

We express our sincere gratitude to the Area Chair for their hard work. We are encouraged that the reviewers recognized the **novelty of our approach** (Reviewers 8Zww, sxvq), the **solid empirical foundation of our motivating study** (Reviewers d5R9, yyBg), and the **substantial performance improvements** (Reviewers yyBg, 8Zww) achieved by our framework.

Below is a summary of our response to the key issues for quick review.

**1. Clarifying the Rationale: From Empirical Observation to Robust Formulation**

To address concerns that our motivation was "heuristic" (Reviewers sxvq, d5R9, 8Zww), we have provided deeper insights and validations to bridge the gap between empirical observation and method design:

-  **Theoretical Grounding:** We clarified that our method works by **filling representational gaps** in the high-dimensional feature manifold, which specifically targets regions where data is "sparse" in diversity rather than just "rare" in count. This moves beyond the "frequency trap" to a rigorous definition of data value.

-   **Component Necessity:** We added a detailed decomposition of the Representation Score (Table 8), proving that **$\mathcal{D}_{vis}$ (Visual Diversity)** and **$\mathcal{D}_{ctx}$ (Contextual Diversity)** address specific failure modes (e.g., large objects and outer regions) that frequency-based methods fail to solve.

-   **Parameter Robustness:** To address the concern that our weighting is heuristic, we provided a sensitivity analysis for $\beta$ (Appendix Table S3), demonstrating that the framework remains stable and effective across different hyperparameter settings. This renders it as a robust solution rather than a tuned heuristic.

**2. Major New Experiments: Generalizability to Modern Detection Paradigms**

To address concerns regarding the applicability of our proposed method to stronger baselines (Reviewer d5R9), we extended our experiments to **YOLOX, DINO, and CO-DETR**. As shown in the new Table 9, our framework yields consistent and significant gains (**~2.6–2.8 mAP**) across these diverse detection architectures. This strongly validates that our RS-driven debiasing generalizes robustly beyond standard baselines.

**3. Visual Blueprint Superiority vs. ControlNet**

To address the comparison with ControlNet raised by Reviewers sxvq and d5R9, we have provided a detailed ablation (Table 6) and visual comparison (Appendix Fig. S1). We empirically demonstrate that standard binary maps (ControlNet) suffer from ambiguity in inter-class and intra-class discrimination. By projecting layouts into high-variance RGB space and handling occlusions, our Visual Blueprint reduces the FID score from 28.14 to **15.24** and outperforms ControlNet by **+3.6 mAP** in detection, proving it is a critical innovation rather than a minor variation.

**4. Validated Interdependence and Stability**

In response to Reviewer yyBg's query on the connection between two contributions (i.e., **visual blueprint** and **RS-driven layout recalibration**), we clarify that RS-driven layout recalibration serves to identify **what to generate**, while visual blueprint targets **how to generate** samples faithfully. We further demonstrate (Appendix Table S2) that high-fidelity synthesis with visual blueprints is the foundation for effective debiasing.
Specifically, applying our debiasing strategy to standard generators yields minimal gains compared to using it with our Visual Blueprint.


**5. Enhanced Reproducibility and Clarity**

We have improved the presentation of the manuscript by adding a pipeline visualization (Figure 2), detailed pseudocode for the Visual Blueprint (Algorithm 1), and a computational cost analysis. These additions ensure that our method is transparent and reproducible.

---

> ### Author Response · Authors · 2025-11-30
> **Summary of changes in revised version**
>
> **Presentation Enhancements**:
>
> 1. **Pipeline Figure**: Added a comprehensive figure illustrating the complete workflow (Figure 2).
>
> 2. **Connection between Motivation and Method**: Strengthened the logical transition from the empirical evidence in $\S$ 2 to the formulation of the Representation Score (RS) (Section 3.1).
>
> 3. **RS Quantization**: Clarified the quantization for box size $s$ and position $u$ (Section 3.1).
>
> 4. **Updates to Diagnostic Experiments**: Revised analyses on Conditional Input, Representation Score, and Generalization to Modern Detectors (Section 5.2).
>
> 5. **Blueprint Visualization**: Added a new figure visually comparing standard simple class maps with our Visual Blueprint (Appendix Figure S1).
>
> 6. **Class Distribution Visualization**: Added a new figure comparing original vs. augmented dataset distributions (Appendix Figure S3).
>
> **Additional Experiments and Validations**:
>
> 1.  **Visual Blueprint Ablation**: Added a step-by-step ablation verifying the efficacy of Pixel Canvas, Instance Discrimination, and Overlap Awareness for L2I synthesis (Table 6).
>
> 2.  **Ablation of RS Components**: Added detailed experiments isolating frequency ($D_{freq}$), visual diversity ($D_{vis}$), and contextual diversity ($D_{ctx}$) (Table 8).
>
> 3.  **Generalization to Modern Detectors**: Added experiments on YOLOX, DINO, and CO-DETR, demonstrating consistent gains ($\sim 2.6-2.8$ mAP) across different detection paradigms (Table 9).
>
> 4.  **Interdependence of Contributions**: Added an ablation study demonstrating that high-fidelity synthesis is a prerequisite for effective debiasing (Appendix Table S2).
>
> 5.  **Hyperparameter Sensitivity**: Added ablation studies for the representation score parameter ($\beta$), confirming the method's robustness (Appendix Table S3).
>
> 6.  **Feature Consistency**: Provided statistical evidence showing that Duality-Aware Generative Alignment maintains the consistency between features produced by detectors and generators (Appendix Table S4).
>
> 7.  **Training Stability**: Provided statistical evidence showing that Duality-Aware Generative Alignment reduces training variance (Appendix Table S5).
>
> **Expanded Technical Details**:
>
> 1.  **Pseudocode for Visual Blueprint**: Added Algorithm 1 to Appendix to detail the core rendering logic.
>
>
> 2.  **Computational Cost Analysis**: Added a detailed breakdown of training overhead and resource requirements for each module (Appendix A.5).
>
> 3.  **Quantization Details**: Clarified quantization steps for box size and position in RS calculation (Appendix A.3).
>
> The major revised contents in the revised version are highlighted in $\color{blue}{blue}$.

---

### Meta-Review · Area_Chair_3vzc · 2026-01-06

**Summary:**

This paper received mixed initial review comments. The common concerns from the reviewers are the heuristic design of the RS, and the unclear justification or deeper analysis of the design. Also reviewers felt the focus of the paper is a bit confused. The goal is to improve detector performance, while the main technical contribution comes from the L2I generation. Very few insights on its impact on the detectors are provided, The authors carefully address these concerns in the rebuttal. AC read both the review comments and the rebuttal in details, and noticed most of the concerns are well addressed by clarification on the motivation and intuition, additional ablation studies (like RS parameters, different components of the RS components, and different SOTA detectors etc.), and explanations on the difference between ControlNet baseline and the newly proposed conditions. Considering the improvements on the detector performance are significant enough, and the idea of data synthesis has stronger motivation than naive augmentation, AC suggests to accept the paper as a poster presentation. More details are below.

**Reviewer Concerns:**

AC believes the overall technical concerns are well addressed in the rebuttal, including but not limited to,
- RS parameters ablation
- computational cost
- the effectiveness of the proposed layout blueprint
- gap between real and synthetic data during training
- comparison between freq-aware Gen and the proposed RS
etc.

But the remaining concerns are still outstanding,
- the proposed RS is still too heuristic
- it lacks sufficient detector-centric analysis and deeper understanding

**Reviewer Scores:**

Reviewer 8Zww and reviewer yyBg will possibly remain positive after reading the rebuttal, and improve the scores after seeing the additional ablation studies.
Reviewer sxvq is highly possible to flip the score, given his main concerns have been addressed. The additional ablation studies are not as detailed as step-by-step, but most of the contents have been made up in the revised version.
Reviewer d5R9 might also increase the scores, given the authors made up experiments on other detectors, and ablations on RS parameters are added for clarification.

---

### Decision · Program_Chairs · 2026-01-26

Accept (Poster)